

# Coupling a gas chromatograph simultaneously to a flame ionization detector and chemical ionization mass spectrometer for isomer-resolved measurements of particle-phase organic compounds

Chenyang Bi[1], Jordan E. Krechmer[2], Graham O. Frazier[1], Wen Xu[2], Andrew T. Lambe[2], Megan S. Claflin[2], Brian M. Lerner[2], John T. Jayne[2], Douglas R. Worsnop[2], Manjula R. Canagaratna[2], Gabriel Isaacman-VanWertz[1]

[1]Department of Civil and Environmental Engineering, Virginia Tech, Blacksburg, Virginia, 24060, USA
[2]Aerodyne Research Inc, Billerica, Massachusetts, 01821, USA

*Correspondence to*: Gabriel Isaacman-VanWertz (ivw@vt.edu)

**Abstract**

Atmospheric oxidation products of volatile organic compounds consist of thousands of unique chemicals that have distinctly different physical and chemical properties depending on their detailed structures and functional groups. Measurement techniques that can achieve molecular characterizations with details down to functional groups (i.e., isomer-resolved resolution)

are consequently necessary to provide understandings of differences of fate and transport within isomers produced in the oxidation process. We demonstrate a new instrument coupling the thermal desorption aerosol gas chromatograph (TAG), which enables the separation of isomers, with the high-resolution time-of-flight chemical ionization mass spectrometer (HR-ToF-CIMS), which has the capability of classifying unknown compounds by their molecular formulas, and the flame ion detector (FID), which provide a near-universal response to organic compounds. The TAG-CIMS/FID is used to provide

isomer-resolved measurements of samples from liquid standard injections and particle-phase organics generated in oxidation flow reactors. By coupling a TAG to a CIMS, the CIMS is enhanced with an additional dimension of information (resolution of individual molecules) at the cost of time resolution (i.e., one sample per hour instead of per minute). We found that isomers are prevalent in sample matrix with an average number of three to five isomers per formula depending on the precursors in the oxidation experiments. Additionally, a multi-reagent ionization mode is investigated in which both zero air and iodide are

introduced as reagent ions, to examine the feasibility of extending the use of an individual CIMS to a broader range of analytes with still selective reagent ions. While this approach reduces iodide-adduct ions by a factor of two, $[M-H]^-$ and $[M+O_2]^-$ ions produced from lower-polarity compounds increase by a factor of five to ten, improving their detection by CIMS. The method expands the range of detected chemical species by using two chemical ionization reagents simultaneously, enabled by the pre-separation of analyte molecules before ionization.



## 1 Introduction


Atmospheric aerosols can impact climate by scattering light and absorbing radiation thus changing earth's reflectivity directly, or by impacting the formation of clouds (Seinfeld and Pandis, 2016). If inhaled, they may deposit into the alveoli region of lungs and transmit into the blood to cause adverse health effects such as ischemic heart disease, cerebrovascular disease, and lung cancer (Burnett et al., 2014; Pope and Dockery, 2006). A significant fraction of sub-micrometer aerosol mass is organic,

mostly formed through the oxidative conversion of gas-phase volatile organic compounds (VOCs) to lower-volatility oxygenated organic compounds that can condense to form secondary organic aerosol (SOA) (Goldstein and Galbally, 2007; Jimenez et al., 2009; Kroll and Seinfeld, 2008).

Atmospheric oxidation of volatile organic compounds produces thousands of unique chemicals that exist in a dynamic and

complex mixture that span a wide range of physicochemical properties (Atkinson and Arey, 2003; Goldstein and Galbally, 2007). The detailed structure and functional groups of each individual compound controls its properties (Kroll and Seinfeld, 2008; Nozière et al., 2015). Rate of reactions, partitioning between phases, and compound toxicity are all impacted by the functional groups present in a molecule (Arangio et al., 2016), and in some cases on its physical conformation (Atkinson, 2000; Lim and Ziemann, 2009). Characterization of a compound to the level of detail of its molecular structure (i.e., isomer-resolved

composition) is therefore necessary to predict whether a compound will deposit to a particle or surface, react with an oxidant, or persist in the atmosphere, and to understand its potential adverse impacts on public or ecosystem health.

A major advance in the online non-target analysis of oxygenated organics in the atmosphere has been the development and widespread adoption of the high-resolution time-of-flight chemical ionization mass spectrometer (HR-ToF-CIMS). These

systems directly sample atmospheric constituents with minimal pre-treatment and classify them by their molecular formulas, providing unprecedented characterizations of thermally and/or chemically labile atmospheric constituents. Multiple negative-ion chemical ionization methods have been applied to atmospheric sampling, including acetate for the detection of carboxylic acids and some inorganic acids (Bertram et al., 2011; Brophy and Farmer, 2016; Veres et al., 2008), nitrate for the detection of highly oxidized organic matters (Ehn et al., 2010; Krechmer et al., 2015), and iodide for the detection of moderately and

highly polar organic compounds (Aljawhary et al., 2013; Isaacman-Vanwertz et al., 2018; Lee et al., 2014; Lopez-Hilfiker et al., 2014; Riva et al., 2019). The latter has seen particularly wide use due to its relatively general selectivity and straightforward chemistry, in which an adduct is formed between the analyte and iodide ion. Due to the high negative mass defect of iodine, iodide adduct ions are separated from other isobaric ions in the mass spectrum by the so-called "iodide valley" in which few ions are present, so can be classified by their molecular formula with high confidence and low detection limits (Lee et al.,

2014). Ions with positive mass defects (i.e., non-iodide-adduct) are observed in iodide-CIMS mass spectra of atmospheric mixtures and may represent peroxy acids (Lee et al., 2014; Mielke and Osthoff, 2012), but these ions are generally poorly understood and frequently excluded from analyses due to their uncertain interpretation.



A major barrier to understanding most reagent ion chemistries is that isomers cannot be differentiated since analytes are classified by molecular formulas. Studying instrument response requires introducing analytes as known standards, so is limited to commercially available or custom synthesized compounds. Such compounds do not fully capture the composition of real-world atmospheres, which contain compounds that are not easily synthesized or available, such as transient and short-lived oxidation products. An improved approach would be the generation of oxidized products through the simulated atmospheric oxidation of atmospherically-relevant precursors, but this approach has limited utility due to the formation of multiple isomers of each formula (Brophy and Farmer, 2016). The range of CIMS reagent ion chemistries that are practically useful is therefore limited to those, such as iodide, that yield distinguishable products, while other potentially promising techniques (e.g., acetate CIMS) have seen less use due to the complex ionization pathways. The lack of isomer-resolution afforded by CIMS introduces several other limitations as well. The fate of a compound and its impacts on the environment are dependent on molecular structure, so they may not be fully predicted or captured by molecular formulas. Furthermore, a formula may represent many isomers, but CIMS signals will preferentially represent isomers to which the ionization method is more sensitive. This may bias the interpretation and quantification of CIMS data, though the extent to which this issue has impacted the literature is unknown.

The resolution of individual components in a complex mixture is typically performed through some form of chromatography. While liquid (e.g., Gómez-González et al., 2008), ion (e.g., VandenBoer et al., 2012), and gas chromatography (e.g., Hamilton, 2010; Schauer et al., 1996) have all been used in atmospheric measurements, we limit our discussion here to gas chromatography (GC), as it has been demonstrated as a field-deployable technique for the analysis of oxygenated organics, complementary with CIMS instrumentation (Isaacman-VanWertz et al., 2016; Thompson et al., 2017; Zhang et al., 2018). GC separates compounds in a complex mixture based upon their vapor pressure and polarity and has been field-deployed for decades for the online measurement of low-polarity gas-phase components (Goldan et al., 2004; Goldstein et al., 1995; Helmig et al., 2007; Millet, 2005; Prinn et al., 2000; Vasquez et al., 2018). More recent developments on the field-deployable thermal desorption aerosol gas chromatograph (TAG) have enabled the GC analysis of semi-volatile and particle-phase air samples, through novel sampling cells, autonomous calibration via liquid injections of authentic standards, and derivatization of oxygenates for the analysis of oxygenated compounds (Isaacman et al., 2014; Williams et al., 2006; Zhao et al., 2013). Detection of analytes eluting from the GC may rely on a flame ionization detector (FID), which has near-universal response but provides no chemical information about an analyte (Kolb et al., 1977; Willmott, 1978), or an electron ionization mass spectrometer (EI-MS), which provides identification of compounds with mass spectra available in existing libraries but requiring careful interpretation to identify those not in the libraries. These techniques have allowed quantification of some individual isomers in SOA, but the substantial majority of analytes observed in GC analyses of SOA have no known structures or identification (Isaacman-VanWertz et al., 2016) due to a lack of commercially available standards of atmospheric oxidation products.



We demonstrate here a new instrument coupling the TAG, which enables the separation of isomers, with the HR-ToF-CIMS, which has the capability of classifying unknown compounds by their molecular formulas, and the FID, which provide a near-universal response to all analytes for quantification with low uncertainty. The specific objectives of this work are to 1) describe the set-up and operation of a TAG-CIMS/FID; 2) evaluate the presence of isomers in laboratory-generated SOA; 3) demonstrate the utility of this instrument for investigating ionization chemistries by resolving mass spectra of individual compounds separated from a complex mixture; and 4) expand the range of chemical species which can be measured by an individual CIMS by using multiple reagent ions.

## 2 Instrumentation and methods

As shown in Figure S1, the TAG-CIMS/FID integrates the use of a GC instrument, TAG, with two detectors, HR-ToF-CIMS (Aerodyne Research Inc.) and FID (Agilent 7820A). The GC column effluent is split to the two detectors using a heated and passivated tee together with heated fused-silica transfer lines. The instrument allows the collection of chromatographic signals generated by CIMS and FID simultaneously.

### 2.1 HR-ToF-CIMS

The HR-ToF-CIMS (Lee et al., 2014) uses a similar physical configuration as described by Isaacman-Vanwertz et al., (2018) with modification at the inlet for adapting the GC flow. The inlet is a heated metal cartridge which is kept at 225°C and has a 1/32" inner diameter bore-through center hole to allow insertion of fused-silica guard column into the ion-molecule region (IMR). The tip of the guard column extends 2 mm out of the heated cartridge to mix with the reagent ion flow inside the IMR.

Two different ionization modes are used in this study: iodide and multi-reagent ionization mode. The iodide ionization mode (Lee et al., 2014) is used in a similar configuration to those published in the past for a gas-phase iodide CIMS (Krechmer et al., 2016). Iodide ions are generated by passing a 2 slpm flow of humidified ultrahigh purity (UHP) $N_2$ over a permeation tube filled with methyl iodide and then through a radioactive source (Po-210, 10 mCi, NRD) into the IMR. The IMR pressure is maintained at 100 mbar. Besides the pure iodide ionization mode, the CIMS operated using a multi-reagent ionization mode to investigate the use of multiple reagent ions in the IMR. Instead of using pure nitrogen, the multi-reagent ionization mode uses a mixture of 1.91 slpm of UHP $N_2$ and 0.09 slpm of ultra-zero air as the methyl iodide sheath flow. This mixture is selected such that iodide ions ($I^-$) accounted for ~50% of the total reagent ion count, with the rest comprised of air-based components as described later.

To obtain a smooth chromatographic peak, raw negative-ion spectra are typically acquired at a rate of 4 Hz without time averaging, which is higher than the rate typically used for direct air sampling by CI-TOF-MS (1 Hz and 1 min averaging).





Data collection as fast as 20 Hz was tested and found to be viable for applications requiring higher data frequency, but is not necessary for data collection described here. The ToF-MS is operated in "V-mode" and achieves a mass resolution of ~4100 at m/z 212 and a mass accuracy of <5 ppm, which enables the assignment of elemental composition to observed mass-to-charge values.

## 2.2 TAG

The TAG used in this study is a custom-built instrument similar to those commercially available through Aerodyne Research Inc. In brief, aerosols are collected with a sampling flow rate of 9 slpm through a 9-jet impactor with a 50% collection cut-point of 0.085 μm (Williams et al., 2006). The collection cell is equipped with a liquid injection port where chemical standards can be quantitatively added through an automatic liquid injection system (Isaacman et al., 2011). After a 5-15 mins sample collection period, which is equivalent to 0.045-0.135 m$^3$ of air sampled, samples are thermally desorbed in helium to the head of the GC column by heating the cell to 300 °C at a rate of 50 °C /min while the GC column is held at 50 °C with ~ 3 sccm of helium carrier gas flow. Samples pass through a passivated metal manifold with restrictive capillaries to separate sample pressures from GC operating pressures without the use of in-line valves (Kreisberg et al., 2014). A polar GC column (MXT-WAX, 17 m × 0.25 mm × 0.25 μm, Restek) wrapped onto the temperature-controlled metal hub is used for the separation of oxygenated organic compounds. The GC analysis is conducted with 1 sccm of helium and the temperature of the GC column is ramped to 250 °C at a rate of 10 °C /min, then held for 25 mins. Though analysis focuses on oxygenated compounds to which the CIMS may exhibit sensitivity, we highlight that derivatization, which is typically used in the analysis of oxygenates by GC (Isaacman et al., 2014), is not employed here to minimize chemical alterations to the functionality of the analytes reaching the detectors. Although not observed with the injection of liquid standards, it is possible that skipping derivatization may bring up certain limitations of the instruments such as the decrease of the transmission efficiency and stability of the analytes (Isaacman et al., 2014).



## 2.3 TAG-CIMS/FID interface

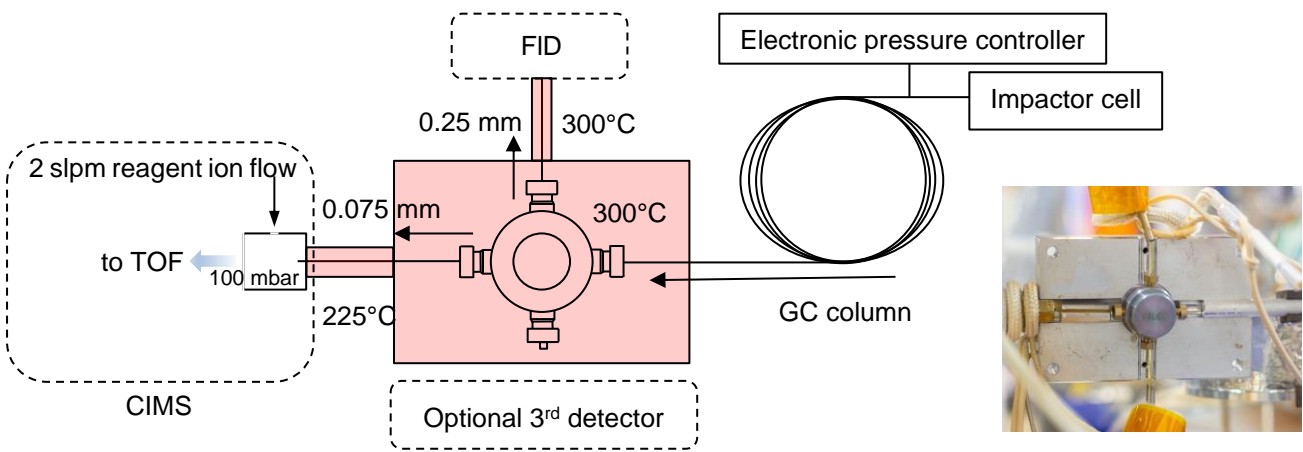

**Figure 1.** Schematics of the TAG-CIMS/FID interface, with a photo inserted. Details of each instrument are simplified. Restrictive capillaries to each detector are used to balance flows to each detector due to differences in pressure (dimensions and temperatures shown).

The interface between the GC and detectors controls the relative transfer flow toward each detector and must avoid any degradation to the chromatography (i.e., cold spots or dead volume). As shown in Figure 1, a 1/32" passivated (SilcoNert 2000, SilcoTek Corp.) metal cross (Part No. ZX.5, VICI) is covered by a two-piece heated aluminum block. The four ports on this connector allow splits from the GC to three detectors (FID, CIMS, optional third). Because the IMR pressure of the CIMS is at 100 mbar while the FID is at ambient pressure, the restriction to the CIMS side needs to be significantly higher than that of

the FID side to maintain comparable flow rates. These restrictions are achieved using deactivated fused-silica guard columns which have an inner diameter of 0.075 and 0.25 mm, and a length of 0.18 and 0.5 m to connect the cross with CIMS and FID, respectively. Because the change of temperature of GC column during a run cycle may potentially influence the split flow, we mitigate the impacts by maintaining the TAG-CIMS/FID interface at a constant temperature, 300 °C, that is 50 °C higher than the maximum temperature of the GC column. The guard column to FID passes through a narrow-bore metal tube wrapped in

heater cable with fiberglass insulation and heated to 300 °C to prevent the condensation of analytes in the transfer lines. The temperature of the entrance capillary to the IMR of CIMS is maintained at 225 °C in order to prevent the degradation of PTFE components of CIMS inlet; this leads to some peak broadening of low-volatility analytes in the CIMS data. With these dimensions and temperatures, the flow rate to FID is approximately one-third of total GC flow (0.3 sccm, measured using Sensidyne Gilibrator-2 at the inlet of FID) with the remainder to the CIMS (0.7 sccm). To further evaluate the stability of the

split ratio of flow, test runs were conducted prior to the experiments to monitor the flow rate at the inlet of FID, variability in the flow split was found to be less than 10% variation throughout a run cycle, stable enough to be quantitative. Additionally, the CIMS was swapped with an EI-MS while maintaining the TAG-MS/FID interface so that liquid injections of alkanes standards (i.e., alkanes mix C8-C40, AccuStandard) can be measured by both EI-MS and FID. The ratios of EI-MS to FID



peak area for observed alkanes, which linearly correlate with the flow split ratios at a given retention time, were similarly
found to vary by less than 10%.

## 2.4 Laboratory experiments

Data are collected from two sources: injection of liquid standards, and laboratory-generated SOA through the oxidation of
atmospherically relevant precursors. Liquid standards include 1,12-dodecanediol (Sigma Aldrich, 99% purity), undecanoic
acid (AccuStandard, 100% purity), eugenol (Sigma Aldrich, 99% purity), vanillin (Sigma Aldrich, 99% purity), vanillic acid
(Sigma Aldrich, 97% purity), and oleic acid (Sigma Aldrich, 99% purity). Additionally, six commercially available liquid
fragrance samples (MakingCosmetics Inc) are also used to serve as representative complex mixtures. SOA was generated via
gas-phase $O_3$ and/or OH oxidation of limonene (Sigma Aldrich, 97% purity), 1,3,5-trimethylbenzene (Sigma Aldrich, 98%
purity), and eucalyptol (Sigma Aldrich, 99% purity) in a Potential Aerosol Mass (PAM) flow reactor (Lambe et al., (2011),
and collected onto the TAG impactor cell at 9 L/min sampling flow rate, and analyzed by the TAG-CIMS/FID. We hereafter
label data from a given set of oxidation experiments system as "precursor-oxidant" (e.g. limonene-OH).

## 2.5 Data processing

CIMS and FID data are acquired as continuous timeseries throughout a single GC run. TAG sends a start signal to both CIMS
and FID once the GC run starts and sends a digital signal to stop the data acquisition once a sample run is completed. For each
run file, high-resolution (HR) mass spectrum peak fitting of CIMS data is conducted using the Tofware (Tofwerk, AG and
Aerodyne Research, Inc., version 3.1.2) toolkit developed for the Igor Pro 7 analysis software package (Wavemetrics, Inc.).
HR fitting of iodide-adduct ions was used to generate a list of potential molecular formulas present within a given set of
experiments. Once the formulas are extracted and assigned in CIMS, the high-resolution time-series data with the peak list are
then imported into TERN, the freely-available (https://sites.google.com/site/terninigor/) Igor-based software tool for the
quantification of chromatographic data (Isaacman-VanWertz et al., 2017). The retention times of peaks between CIMS and
FID are aligned based on linear regression of retention times of internal standards (i.e., vanillin and 1,12-dodecandiol) and the
most abundant compounds in chromatographic runs.



# 3 Results and discussion

## 3.1 Chromatograms

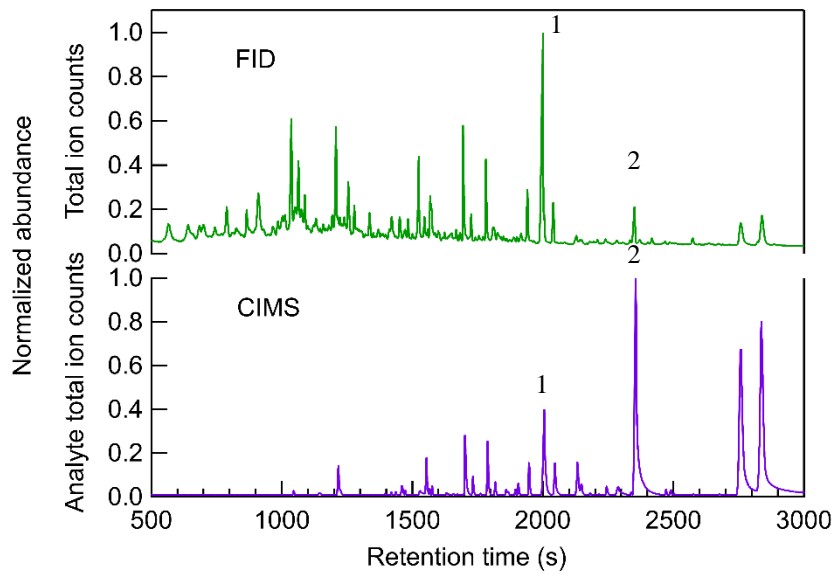

**Figure 2.** Comparison of chromatograms between FID and CIMS from particle sample generated from limonene-$O_3$ reaction. The time-series signals of CIMS are the analyte total ion counts, which are summation of all ions with the reagent ions removed while signals of FID are single-channel. The signals of both CIMS and FID are normalized to the highest peak in the chromatogram.

Figure 2 shows an example comparison of chromatograms collected by the FID and CIMS for limonene-$O_3$ SOA. The FID produces a total signal (y-axis) as a function primarily of the mass of carbon combusted, so each analyte (i.e., chromatographic peak eluting at a given retention time) responds with similar mass-based sensitivity. However, the total chromatogram shown is the full extent of the data available for a given sample; there is no further separation by mass or other dimension. Co-eluting peaks that are not well resolved may not be able to be accurately integrated. In contrast, the CIMS analyte total ion count signal

is the sum of all observed ions with the reagent ions removed, so while two peaks may not be chromatographically resolved (i.e., co-eluting total signal), they may be mass-resolved by producing signals on different ions. The results demonstrate that the responses from FID and CIMS have distinctly different patterns with the number of peaks and the height of aligned peaks in the chromatograms of FID and CIMS differ significantly. Many early-eluting compounds (i.e., compounds having peaks with retention times less than 1500 seconds) that are observed by FID do not produce a clear signal in the CIMS. Compounds

with retention time longer than 1500 seconds usually have peaks detected by both FID and CIMS chromatograms. Since the TAG-CIMS/FID interface and the capillary to the FID is held at 50 °C above the maximum column temperature, differences in the transfer of analytes to these two detectors should be negligible. Instead, these differences are due to the selectivity of the two detectors. FID is a near-universal detector, able to detect almost all organic compounds with relatively similar and predictable responses (Scanlon and Willis, 1985). The sensitivity of iodide-CIMS is highest for compounds that are more polar



and can therefore more readily form an adduct with the iodide ion (Iyer et al., 2016). The wax GC column used here more readily retains polar compounds, suggesting that the early-eluting analytes are more likely to be lower-polarity compounds that exhibit low sensitivity in the iodide-CIMS.

The low abundance of early-eluting peaks in the CIMS chromatogram is an example of the value of the coupled CIMS/FID

detector pair for investigating CIMS response. Although compounds having larger retention times can be found in both FID and CIMS, their abundance is significantly different between the two detectors. For example, Analyte 1 ($C_{10}H_{14}IO_3^-$) with a retention time of 2000 seconds has the most abundant signals in FID while Analyte 2 ($C_8H_{10}IO_4^-$) with a retention time of 2380 seconds is the largest peak in CIMS. Since FID sensitivity differs between compounds by less than a factor of two (Hurley et al., 2020), the peak area of FID approximates the relative mass of a compound in a sample matrix. However, comparing to the

near-universal response of FID signals, the signals of iodide CIMS per unit mole of analytes may vary up to five orders of magnitudes and highly depend on their enthalpies of binding with iodide (Iyer et al., 2016; Lopez-Hilfiker et al., 2016a). The two peaks highlighted provide an example in the variability of CIMS response: Analyte 1 has a larger FID peak area, indicating a higher mass concentration in the sample mixture than Analyte 2. However, since the CIMS peak area of Analyte 1 is lower, it must be less sensitive than Analyte 2 in an iodide CIMS. With the use of FID in addition to the CIMS detector, calibration

of compounds in CIMS without using authentic standards can therefore theoretically be achieved. Implementation of this calibration approach including detailed methods of quantification and determination of isomer sensitivity is complex and will be addressed in future work. This manuscript focuses instead on the description of TAG-CIMS/FID, isomer-counting, and evaluation of multi-reagent CIMS.

**3.2 Number of isomers per formula**

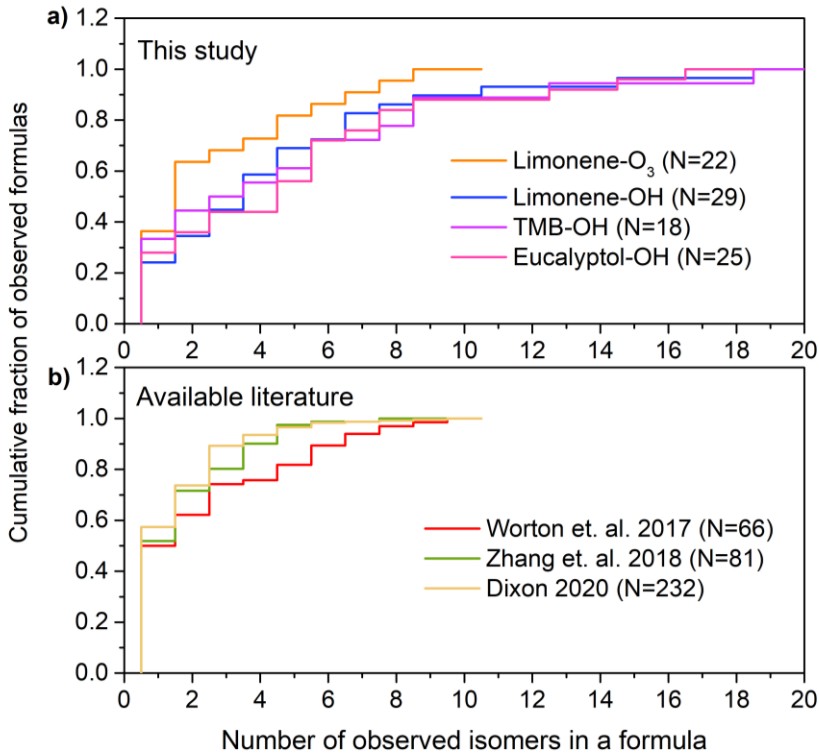

**Figure 3.** Number of isomers per chemical formula in (a) observed in this work in oxidation experiments, and (b) available literature. Data in (b) are from ambient atmospheres in a ponderosa pine forest in California measured by two-dimensional GC without derivatization (Worton et al., 2017), Southeastern U.S. by two-dimensional GC with derivatization (Zhang et al., 2018), and combined datasets from SOAs formation measured by liquid chromatography (Bryant et al., 2019; Dixon, 2020). The number, N, of formulas observed in each dataset is provided in the legend.

We explore here the extent to which isomers are present in atmospherically relevant mixtures to assess the potential utility of a TAG-CIMS/FID (or, more generally, a GC-CIMS/FID). The number of isomers per molecular formula identified in the high mass resolution peak fitting can be obtained by counting the number of peaks in the chromatograms of a specific high-resolution *m/z* of the iodide-adduct (i.e., a specific molecular formula). These data should be interpreted cautiously.

Overestimation may occur because peaks observed might be formed in part by thermal decomposition of analytes during thermal desorption. Conversely, isomers may be undercounted if they fall below the threshold used to count them, which was selected here to be conservative. The limitations in these estimates will be discussed in detail later in this section, but these data nevertheless provide a useful, if uncertain, understanding of the prevalence of isomers in atmospherically-relevant samples.






Figure 3a shows the cumulative number of isomers found in formulas identified in the PAM oxidation experiments. In this study, compounds with a peak height higher than $1.0 \times 10^4$ ions/s are taken into the isomer counts; the number of isomers is sensitive to the selection of this threshold so we set as a threshold the approximate level at which the chromatographic peak height clearly rises above the baseline by at least a factor of 50; in many cases, peaks are present below this threshold (e.g.,
Figure S2, showing 12 isomers identified in $(C_9H_{12}O_4)I$). Although 30% of formulas in the oxidation experiments have only one isomer, a significant portion (34%) of formulas have more than five isomers. For limonene-$O_3$, limonene-OH, 1,3,5 trimethylbenzene-OH, and eucalyptol-OH reactions, the median number of isomers per formula is 2.0, 4.0, 3.5, and 5.0; the average number of isomers per formula is 3.7, 4.9, 5.1, and 5.3; and the maximum number of isomers per formula is 14, 19, 19 and, 17, respectively. The results indicate that isomers are prevalent in sample matrix with an average number of three to
five isomers per formula depending on the precursors in the oxidation experiments. We compare these data to previously published studies using isomer-resolved analyses of SOA (Figure 3b). Ambient measurements from a range of environments show a qualitatively similar distribution of isomers per formula, though with somewhat lower averages of 2.7, 2.1, and 1.9, based on data collected from ambient air in a ponderosa pine forest in California measured by two-dimensional GC without derivatization (Worton et al., 2017), Southeastern U.S. by two-dimensional GC with derivatization (Zhang et al., 2018), and
combined datasets from SOA formation measured by liquid chromatography (Bryant et al., 2019; Dixon, 2020). These data were collected by different instruments (two-dimensional GC with filter samples analyzed by thermal-desorption and derivatization, two-dimensional GC with filter samples analyzed by thermal desorption, and LC, respectively) that are likely not sensitive to the same compounds. Another possible reason for the lower number of isomers might be that the use of two-dimensional GC can limit the range of compounds detected since those analytes have to be such that they make it through two
consecutive capillary columns. Together, the published data and that collected by TAG-CIMS/FID support the conclusion that isomers are abundant in atmospheric samples. Those isomers may have significantly different physical and chemical properties that impact the formation, transport, and toxicity of SOA, and the distribution of isomers could vary temporally or spatially. The isomer-resolved classification of SOA components provided by TAG-CIMS/FID therefore provides valuable understandings of the oxidative process. Although the molecular structure of each isomer cannot be recognized directly in the
chromatograph and the specific functional groups within the formula remain unknown, the chromatographic separation of the TAG-CIMS provides some comparison of polarity, which is dependent on chemical structures, for isomers within a formula.

There are certain limitations for the analysis of the number of isomers per formula in this study. The number of isomers is probably underestimated due to exclusion of peak heights less than $1.0 \times 10^4$ ions/s. Furthermore, isomers may vary in their
sensitivity, with isomers having less polar functional groups possibly not detected by CIMS. Additionally, the use of a GC column, which is selective towards a certain range of volatility and polarity of compounds, limits the detection of compounds. Conversely, thermal desorption may fragment larger accretion products to form analytes not present in the original sample (Isaacman-VanWertz et al., 2016; Lopez-Hilfiker et al., 2016b), or may reverse particle-phase oligomerization reactions (Claflin and Ziemann, 2019). These fragments may be identified as oxidation products in this analysis and consequently



overestimate the number of isomers. We note, however, that similar numbers of isomers are observed when using liquid
chromatography (Figure 3b), which does not involve thermal desorption. Given these uncertainties, we believe that the results
presented are not a floor or a ceiling on the number of isomers in the atmosphere, but a step toward understanding a poorly
constrained problem. In any case, the number of isomers observed by any single instrument is expected to be underestimated
as no instrument is capable of measuring all atmospheric components with molecular speciation. The number of isomers shown

in Figure 3 is therefore likely more illustrative as an example than quantitative, demonstrating the general ubiquity of isomers
in the atmosphere. This issue raises significant questions the atmospheric measurement community should address as to how
isomers differ in their sources, physicochemical properties, instrument sensitivities, and atmospheric transformations.

## 3.3 Iodide ionization versus multi-reagent ionization

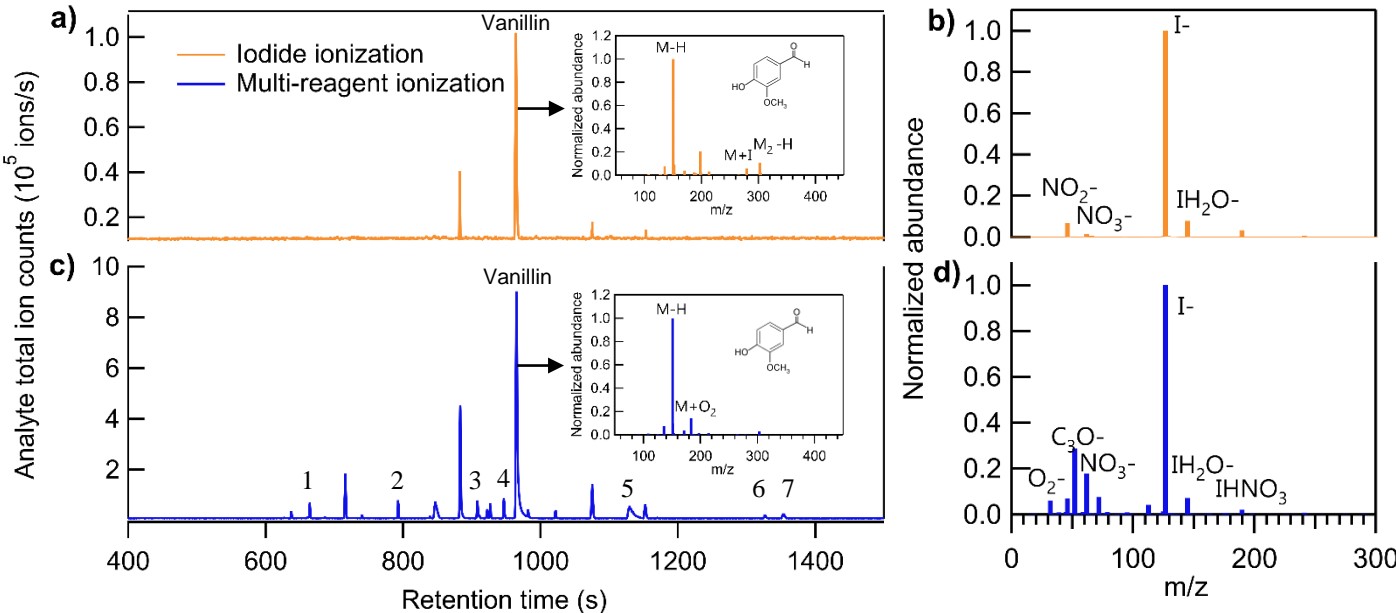

**Figure 4.** Comparison of chromatograms of analyte total ion counts between a) iodide ionization and c) multi-reagent ionization. Comparison of background mass spectra between b) iodide ionization and d) multi-reagent ionization.. The sample introduced in this run is the mixture of liquid chemical standards and six commercially available fragrances.

Unlike direct air sampling by CIMS, in which the mass spectrum at a given time point is the summation of all analytes, the

mass spectra of TAG-CIMS/FID analytes are separated in time by chromatography. Consequently, if a chromatographic peak
of a compound is well-resolved in CIMS, all signals detected from the ionization of a single analyte are observed at the same
chromatographic retention time and unambiguously assignable to that specific compound, including iodide adducts, products
of adduct declustering, fragments (generally not from iodide clustering), and any ions produced by simultaneous alternate
chemistry with other ions present in the atmospheric pressure interface (e.g., air). This provides a clean mass spectrum for





each chromatographically well-resolved analyte and consequently a significant advantage for understanding ionization

chemistry. Previous work has demonstrated that coupling a GC-interface to a $NO^+$ CIMS can determine the products ion

distributions for VOCs (Koss et al., 2016). This instrument complements this previous work by examining less volatile and

more oxidized compounds, as well as other CIMS chemistries. This technique is particularly interesting in the context of iodide

CIMS chemistry, as it allows us to explore ions with positive mass defects (i.e., non-adduct ions), which are not particularly

well understood (Lee et al., 2014). We demonstrate the capability of the technique by showing the chromatograms of a complex

sample containing a mixture of liquid chemical standards and six commercially available fragrances in Figure 4. For example,

while the analyte mass spectrum of vanillin, shown in Figure 4a, in iodide ionization mode does contain an iodide-adduct ion

(i.e., $[M+I]^-$), there are other ions with higher abundance including the predominant deprotonated form of vanillin (i.e., $[M-H]^-$

), followed by its nitrite-adduct (i.e., $[M+NO_2]^-$), then the deprotonated form of dimer (i.e., $[M_2-H]^-$). In other words, this

compound, which is generally measurable by iodide-CIMS (Gaston et al., 2016), produces a large number of detectable ions

through reactions with other reagents in the IMR. Similar trends are observed for other compounds injected as authentic

standards, including undecanoic acid and 1,12-dodecanediol. In contrast, the iodide-adduct ions ($[M+I]^-$) of more polar and

lower-volatility aerosol constituents produced in oxidation experiments are the dominant ions in their analyte mass spectra.

Application of this instrument to ambient samples and/or selected test systems have therefore be a pathway toward better

understanding iodide adduct chemistry as and co-existing side reactions.

Using GC-CIMS not only enables the elucidation of different ionization pathways in the CI source and enables separation of

interferences in the quantification, but might also be useful for exploiting these co-existing chemistries to yield additional

information. While chemical ionization intrinsically offers selectivity for ease of analysis, selectivity is also negatively limiting

(Munson and Field, 1966). Thus, under certain circumstances it may be useful to use multiple reagent ions to detect different

classes of compounds using separate, but still soft, ionization methods. Other ions like the deprotonated form of the analyte,

$[M-H]^-$ spectra might be better suited for the identification and quantification of some analytes. The deprotonated ions are

believed to be produced through the reaction with $O_2^-$ present in the IMR (Dzidic et al., 1975; Hunt et al., 1975). It is reported

that the presence of $O_2^-$, which is commonly found in atmospheric pressure ion sources such as electrospray ionization (ESI)

(Hassan et al., 2017), atmospheric pressure chemical ionization (APCI) (McEwen and Larsen, 2009), atmospheric pressure

photoionization (APPI) (Song et al., 2007), and direct analysis in real-time (DART) (Cody et al., 2005), may result in the

deprotonated molecules through oxidative ionization. Therefore, using multiple reagent ions including $I^-$ and $O_2^-$, it is possible

that low polarity compounds tend to be ionized through proton abstraction by $O_2^-$ while compounds with high polarity can still

form iodide adducts. Although we did not find carbonate (as $CO_2^-$ and $CO_3^-$) in the mass spectrum in this study, we caution

that those ions, like $O_2^-$, can also deprotonate molecules and may interfere with the quantification of the deprotonated ions.

To explore the feasibility of using multiple simultaneous chemistries (e.g., deprotonation reactions and iodide adduct

formation) to extend the utility of a CIMS with isomer resolution, the CIMS was operated in a multi-reagent ionization mode



by adding 5% ultra-zero air to the ionization region alongside iodide. Figure 4b and 4d show the background ions under the

two modes. With no sample introduced into the system (i.e., pure helium as GC effluent), the total ion counts are $1.4\times10^6$ and $2.4\times10^6$ ions/s and the $I^-$ ion counts are $0.7\times10^6$ and $1.8\times10^6$ ions/s for multi-reagent ionization and iodide ionization, respectively. In other words, by mixing the reagent ion flow with 5% air, the $I^-$ ion reduced by half, while the abundance of additional reagent ions such as $O_2^-$ and $NO_3^-$ increased by approximately an order of magnitude.

As shown in Figure 4c, for compounds that can be detected by iodide ionization, the total number of ions produced by an analyte increased by a factor of five to ten after switching to multi-reagent ionization mode (note that the scale of y-axis in Figure 4c is a factor of ten higher than that in Figure 4a). For example, the analyte total ion counts of vanillin (labeled with an arrow, retention time = 965 secs), has a peak height of $1.0\times10^5$ ions/s in iodide ionization mode while the peak height of vanillin in multi-reagent ionization mode is $8.6\times10^5$ ions/s. This increase in ions is observed to occur almost entirely through

the addition of new chemical pathways. In multi-reagent ionization, the three most abundant ions in the vanillin mass spectrum are deprotonation (i.e., $[M-H]^-$), the cluster with $O_2^-$ (i.e., $[M+O_2]^-$), and the deprotonated dimer (i.e., $[M_2-H]^-$). Because of the presence of oxygen in the reagent ion flow, the abundance of $[M-H]^-$ and $[M+O_2]^-$ is enhanced significantly. Though the $[M+I]^-$ is no longer observed in the spectrum, this is only due to the significant increase in other signals; the actual impact on the iodide adduct formation pathway is minor. To demonstrate, we plot the comparisons of the $[M-H]^-$ and $[M+I]^-$ of vanillin

between the two ionization modes in Figure 5. The peak height of the $[M-H]^-$ ion of vanillin increases by a factor of 10, from

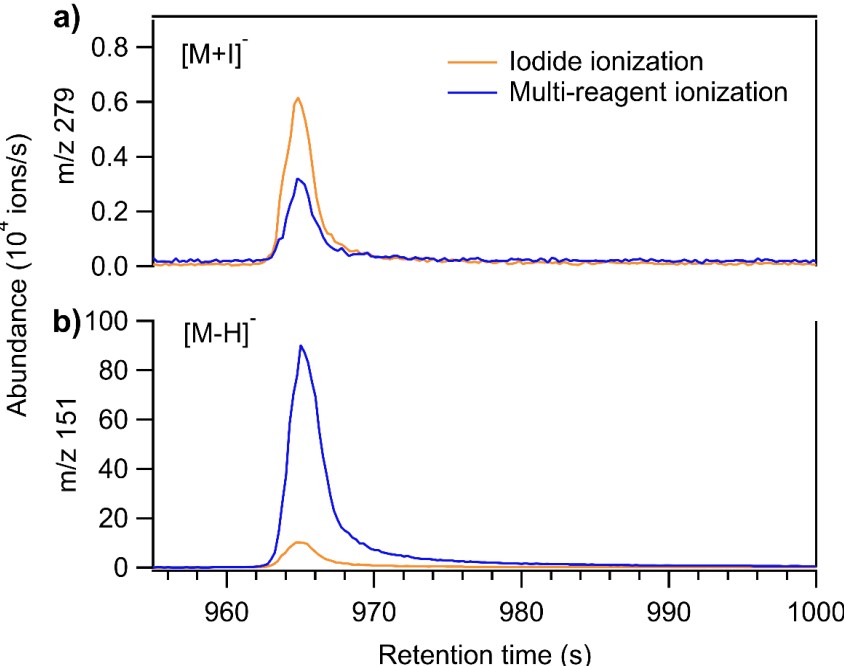

**Figure 5.** Comparisons of chromatograms of a) $[M+I]^-$ ion and b) $[M-H]^-$ ion from the injection of vanillin between iodide ionization and multi-reagent ionization.



$9.0 \times 10^4$ to $90 \times 10^4$ ions/s while the $[M+I]^-$ of vanillin reduces by a factor of only 2, from $0.58 \times 10^4$ to $0.32 \times 10^4$ ions/s after switching from iodide ionization mode to multi-reagent ionization mode, consistent with the factor of 2 decrease in the reagent $I^-$ ion. The results suggest that the instrument selectivity to other classes of compounds can be enlarged by bringing in $O_2^-$ as an additional reagent ion, without significantly suppressing the iodide ionization pathway. In other words, the sensitivity of

compounds that tend to be ionized by $O_2^-$ or other side reactions are significantly enhanced in multi-ionization CIMS with only minor decreases in the sensitivity of compounds typically observed by an iodide-CIMS. As long as individual analytes enter the CIMS at separate times, as in the case of chromatography, combining multiple ionization chemistries can provide additional information or selectivity. An example of the benefit of this approach is demonstrated by the detection of compounds not accessible through iodide adduct formation; 4 times as many compounds are observed in multi-reagent ionization mode (with

formulas assigned to at least half of them). For example, a known component in the sample of complex fragrance mixtures, eugenol (Peak 2 in Figure 4c), is identified in the multi-reagent ionization mode yet not detected in iodide mode. In Figure 4c, 6 other peaks are labeled that are not detected as iodide adducts, but for which formulas can be assigned using $[M-H]^-$ and $[M+O_2]^-$ as identifiers, 1: $C_{15}H_{24}O$, 3: $C_9H_{10}O_3$, 4: $C_{12}H_{24}O_2$, 5: $C_{16}H_{32}O_2$, 6: $C_{18}H_{34}O_2$, and 7: $C_{18}H_{36}O_2$. A reasonable objection to multi-reagent ionization is that the complexity and/or novelty of the chemistry may prohibit reasonable quantification.

However, using CIMS for identification of unknowns by formula or other chemical information is valuable on its own, and quantification of many components is achievable using the FID channel of this instrument. This technique is likely only useful when analytes are individually resolved (i.e., isomer resolution), as the resulting mass spectrum of the complete complex mixture would be otherwise too difficult to interpret.

## 4 Conclusions

We couple a thermal desorption aerosol gas chromatograph with a chemical ionization mass spectrometer as a technique for isomer-resolved analysis of particle-phase organics in the air. The GC column effluent is also split to a flame ionization detector, which provides a near-universal response to carbon-containing analyte, to calibrate the compounds identified by CIMS. We demonstrate that the TAG-CIMS/FID can measure compounds from liquid injections as well as compounds in SOA generated in an oxidation flow reactor. By coupling a TAG to a CIMS, the CIMS is enhanced with an additional dimension of

information (resolution of individual molecules) at the cost of time resolution (i.e., one sample per hour instead of per min). This trade-off may be valuable in ambient atmospheres, as the number of isomers per formula is observed in ambient samples and oxidation experiments to be typically 2 to 5, and as high as 10 to 20, depending on the instrument and the environment.

A key advantage of coupling a TAG to a CIMS is in characterizing the reagent-analyte reactions occurring in the IMR of a

CIMS by resolving mass spectra of individual analytes that might not be commercially available. While the iodide-adduct ions do exist in the mass spectrum of individual analytes, we also observe high abundance of non-adduct ions such as $[M-H]^-$ and $[M+O_2]^-$, which likely account for many ions in the non-adduct region of the iodide valley. By separating analytes





chromatographically, these non-adduct ions can be used for the identification of some compounds. These non-iodide ionization pathways can be further enhanced by the intentional introduction of multiple reagent ions.


A multi-reagent ionization mode is investigated in which both zero air and iodide are introduced as reagent ions, to examine the feasibility of extending the use of an individual CIMS for detection of a broader range of analytes. While this approach reduces iodide-adduct ions by a factor of two, $[M-H]^-$ and $[M+O_2]^-$ ions produced from less polar compounds increase by a factor of five to ten, improving their detection by CIMS. The method expands the range of chemical species, which can be
measured by CIMS without losing the advantage of identifying chemical formula using the iodide adducts. This novel multi-reagent approach is made possible by combining GC and CIMS detection together with co-measurements from FID. The advantage of simultaneously measuring FID signal for isomer-resolved quantification of I-CIMS sensitivity will be discussed in more detail in a forthcoming paper. Thus, taken together, the GC-CIMS/FID instrument not only inherently valuable for its resolution of isomers in complex atmospheric samples, but also for its ability to characterize and calibrate known CIMS
chemistries and to investigate novel and complex chemistries.

**Data availability**

All raw and processed data collected as part of this project are available upon request.

**Author contributions**

CB led hardware design, instrumentation, data collection, and data analysis under the guidance of GIVW. GIVW, JEK, BML, and MRC contributed to the development of the theory of the described approach. GIVW, GOF, JEK, JTJ, and DRW contributed to hardware design and instrumentation. JEK, WX, ATL, MSC contributed to data collection. CB prepared the manuscript with contributions by all authors.


**Competing interests**

The authors declare that they have no conflicts of interest.

**Acknowledgments**

This work is primarily supported by the Alfred P. Sloan Foundation Chemistry of the Indoor Environment Program (P-2018-11129). We would like to thank Dr. David Worton, Prof. Haofei Zhang, Prof. Allen Goldstein, Prof. Jacqueline Hamilton, and Dr. William Dixon for sharing their data, shown in Figure 3.

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
