# Peer review of "Coupling a gas chromatograph simultaneously to a flame ionization detector and chemical ionization mass spectrometer for isomer-resolved measurements of particle-phase organic compounds"

_Atmospheric Measurement Techniques, 2020_

## Referee Comment (RC1) · Anonymous Referee #1 · 8 Sep 2020

Review of "Coupling a gas chromatograph simultaneously to a flame ionization detector and chemical ionization mass spectrometer for isomer-resolved measurements of particle-phase organic compounds" AMT 2020-264 Bi et al.

Summary: The authors describe a new instrument consisting of an aerosol impactor, thermal desorption unit, GC, and split column effluent to an FID detector and an iodide-adduct chemical ionization mass spectrometer. The separate components of this in-

strument have been described previously; the combination of the simultaneous FID and CIMS measurement is novel. The goals of this instrument are to (1) isomerically resolve chemical compounds in an aerosol sample, and (2) determine the sensitivity of the CIMS detector to each chemical species. A second novel instrument technique is also described, in which a combination of air and iodide are used as reagent ions in the CIMS. The purpose of this is to ionize both high- and low-polarity compounds.

Major comments: The manuscript is generally well-written and well-organized. The use of chromatography to interpret CIMS spectra is not especially inventive, and because it is so complicated, this instrument will likely not see widespread use in its current form. However, this is a particularly useful instrument because it addresses two major problems with CIMS in atmospheric science: the spectra may be complicated to interpret, and the sensitivities to individual species are difficult to determine. Used under carefully-controlled conditions in a small number of laboratories, this instrument could provide important reference information to interpret iodide-CIMS measurements from field studies.

The analysis of the number of detected isomers was done very well and it is very useful to see the comparison to other studies.

The inclusion of the air/iodide mixed-reagent-ion technique is interesting, but does not seem to really belong here. It is not even indicated in the title of paper and does not address the motivations mentioned in the introduction. This is better suited to a separate manuscript. If this is retained, it needs to be better explained how the mixed-reagent-ion approach relates to the other capabilities of the instrument. An additional figure or other concrete example should be shown to demonstrate how chemical information can be derived from this technique.

Instead of the mixed-reagent ion description, it would be better to include an assessment of the FID-enabled determination of CIMS sensitivities. This is stated several times as the major benefit of the FID. However there are no data shown to demon-

strate this utility. This is a significant weakness of the manuscript. The manuscript could be published with some minor revisions, but would have a much larger impact if the use of the FID to determine sensitivity is demonstrated.

Specific/minor comments: Section 2.3. It would be helpful to summarize the major technical challenges of CIMS/FID split flow at the top of this section.

Page 6 line 166- 170. I don't understand the purpose of the EI-MS experiment. Please explain.

Page 10 Line 235: Wouldn't decomposition during TD change the parent formula as well?

Page 11 Line 260: This statement should be qualified with the volatility- or carbon-number-range analyzed.

Page 11 lines 264-266: Could you not use the EI-MS and identify the isomers via matching to a NIST library?

Page 14 lines 330-358: Can you provide an example of how this can be used? This seems to subvert the main benefits of CIMS, which are a 1:1 correspondence between molecule and product ion, and the retention of the parent ion (low fragmentation). If the only benefit is that it allows detection of some other compounds with the CIMS, why not just use the FID, which detects everything and with roughly equal sensitivity? The spectra are probably too complicated to interpret without GC-preseparation. Maybe there is some information about the structure of the isomer, given the observed mixed I-/O2- product ions. If so, can you provide some examples? Please comment on the use of this method to interpret the non-adduct ions typically observable in I- CIMS spectra. Can you state anything concrete about the identity of these ions, given the results of your experiments?

Technical corrections: Line 221: "magnitudes" -> "magnitude"

---

## Referee Comment (RC2) · Anonymous Referee #2 · 3 Oct 2020

Review of Bi et al., "Coupling a gas chromatograph simultaneously to a flame ionization detector and chemical ionization mass spectrometer for isomer-resolved measurements of particle-phase organic compounds"

Summary:

The authors describe a new instrument configuration using a TAG column upstream of both an FID and a ToF-CIMS. The goal was to address several current measurement

issues by 1) using the GC to separate isomers in the CIMS spectrum, and 2) using the FID as a means of quantifying unknown compounds in the CIMS spectrum. They also show the simultaneous use of I- and O2- as reagent ions, with the goal of expanding CIMS sensitivity to generally less oxidized compounds. Generally, this is an interesting and well written paper that attempts to address several of the critical issues with CIMS spectrum interpretation of ambient or complex data. Separation of isomers seems be useful (at least for compounds that can make it through a GC column, which will be a subset of what I- usually can measure). This technique may well be very useful (especially for simplified systems), however I don't think the authors have fully illustrated this yet. I have a major issue with some of the methods and interpretations thereof that require major revisions. Mainly, the data shown for iodide ionization shows that the ionization process is different than in typical iodide ionization CIMS setups (at least for some compounds including vanillin). This will diminish the ability to use this instrument to guide interpretation of other typical iodide CIMS measurements. I believe the authors need to address this issue (likely by showing more measurements) before making some of the conclusions drawn here. I also have questions about decomposition, and the utility of interpreting multiple reagent ion spectra.

Main comments:

Line 112: Is there a risk of decomposition of organic analyte molecules at this 225C temperature in the IMR, especially on metal surfaces? I would also be worried about fragmentation at the 300C temps upstream. Previous FIGAERO CIMS research has suggested decomposition of oligomers and/or highly functionalized molecules when heating at even lower temperatures than 300C. Please add some references to this previously observed issue, and also add some discussion somewhere in the manuscript of how decomposition would affect your measurements.

Line 211: Related to my previous comment, how much of these early eluting compounds might be fragmentation products as an artifact of the sampling technique?

[Figure]

Fig 4: Regarding decomposition again, one way to investigate would be to show the FID signal for Fig 4 and maybe for more examples where you are injecting single known compounds. How much signal is in the FID but not the CIMS, i.e., how much fragmentation occurs throughout the sampling process? I think this would be important information for a reader to judge the utility of the technique.

Line 297: This inset in Fig 4a indicates a major issue. In a typical iodide CIMS setup, vanillin would be sampled almost completely at the [M+I]- cluster with iodide, but you're showing that in your instrument it is predominantly sampled at the deprotonated [M-H]-. Therefore, the ionization process is different from a normal iodide CIMS. I think this is a major problem that you need to address before publishing under the pretext that your instrument can be used to help generally interpret iodide CIMS measurements. The most likely answer that I see could be that holding the IMR at 225C is causing the changes? Perhaps at those temperatures (and with metal surfaces?), vanillin becomes a gas phase acid and $C_8H_8O_3 + I- –> C_8H_7O_3- + HI$ proceeds? At line 300, you try to address the [M-H]- by stating that vanillin "produces a large number of detectable ions through reactions with other reagents in the IMR." But, there are no other reagent ions there (except NO2- maybe, but that is often present in typical iodide CIMS spectra and is therefore not the cause), so this statement is probably not accurate. So, please address this issue thoroughly. A first step could be to do the same vanillin injection but hold the IMR at room temp. The vanillin signal may smear, but does is show up at [M+I]-? Again, this seems like a major issue since you're trying to say (eg lines 372-374) that you can use this method to investigate non-iodide clusters separated by the 'iodide valley', but you're apparently also drastically changing the ionization method.

Line 303: Interesting that the lower volatility compounds or more polar compounds appear to have different ionization processes relative to the more volatile or less polar compounds like vanillin. Since one of the advantages of I- ionization has been a more consistent (and single) ionization process for the majority of compounds, do you have any thoughts on how this affects interpretation of the spectra?

Line 318: Since you're seeing the deprotonated ions using both ionization methods (just I-, and both I- and O2-), it seems like it would be really hard to convert the signal in multiple reagent ion mode to mixing ratios as the sum of two ionization processes with variable sensitivities. Especially this would be hard if you have are sampling a complex mixture. It makes me wonder if it's feasible to use both reagent ions at the same time, or if you should instead use them one at a time in series. No doubt that O2- ionization gives you a lot of extra information about the less oxidized/polar compounds that I- can't see. Please discuss this to give the reader confidence that using two reagent ions simultaneously is actually a practical scientific improvement. Best would be to show a quantitative example of a calibration curve using both reagents, but possibly this will be the subject of a future manuscript.

Technical comments:

Fig. 3 Caption: extra 's' after SOA

Line 324: This sentence just needs some clarification. You're either adding zero air to the ionizer alongside methyl iodide, or adding O2- to the ionization region (aka IMR) alongside iodide. Also "ionization region" is ambiguous because that could be the ionizer or the IMR.

Line 326: This was confusing to me because you list the numbers with multi-reagent ion first and iodide second, right after referring to Figs. 4b (iodide) and 4d (multi) in the reversed order. Please reverse the order of listing the numbers in order to stay consistent with the Fig.

---

## Author Comment (AC1) · 5 Nov 2020

The authors would like to thank all the reviewers for reviewing our manuscript and for providing incisive and constructive feedback to help us improve the quality of this paper and to address some issues that required further clarification and discussion. We have made revisions to our original manuscript accordingly. Because the two reviewers have raised similar scientific concerns and discussion points, we have responded

to all reviewers in a combined document. Responses to this Reviewer (Anonymous Reviewer #1) are on pages 1-6

Please also note the supplement to this comment:
https://amt.copernicus.org/preprints/amt-2020-264/amt-2020-264-AC1-supplement.pdf
* * *

---

## Author Comment (AC2) · 5 Nov 2020

The authors would like to thank all the reviewers for reviewing our manuscript and for providing incisive and constructive feedback to help us improve the quality of this paper and to address some issues that required further clarification and discussion. We have made revisions to our original manuscript accordingly. Because the two reviewers have raised similar scientific concerns and discussion points, we have responded

to all reviewers in a combined document. Responses to this Reviewer (Anonymous Reviewer #2) are on pages 7-17

Please also note the supplement to this comment:
https://amt.copernicus.org/preprints/amt-2020-264/amt-2020-264-AC2-supplement.pdf

───────────────────────────

[Figure]

**Supplement:**

The authors would like to thank the editor for managing the peer review process and all the reviewers for reviewing our manuscript and for providing incisive and constructive feedback to help us improve the quality of this paper and to address some issues that required further clarification and discussion. We have made revisions to our original manuscript accordingly. The colorings of text in the reviewer response are:

- Light blue: Original reviewer comments
- Dark blue: Original text in the submitted version of the manuscript. **Bolded sentences** are the text added in the revision while  are the text deleted in the revised manuscript.
- Black: Authors' response to the comments and others.

Note that the line number in the response is based on the revised clean-version manuscript.

Anonymous Referee #1

Summary: The authors describe a new instrument consisting of an aerosol impactor, thermal desorption unit, GC, and split column effluent to an FID detector and an iodide adduct chemical ionization mass spectrometer. The separate components of this instrument have been described previously; the combination of the simultaneous FID and CIMS measurement is novel. The goals of this instrument are to (1) isomerically resolve chemical compounds in an aerosol sample, and (2) determine the sensitivity of the CIMS detector to each chemical species. A second novel instrument technique is also described, in which a combination of air and iodide are used as reagent ions in the CIMS. The purpose of this is to ionize both high- and low-polarity compounds.

Major comments: The manuscript is generally well-written and well-organized. The use of chromatography to interpret CIMS spectra is not especially inventive, and because it is so complicated, this instrument will likely not see widespread use in its current form. However, this is a particularly useful instrument because it addresses two major problems with CIMS in atmospheric science: the spectra may be complicated to interpret, and the sensitivities to individual species are difficult to determine. Used under carefully-controlled conditions in a small number of laboratories, this instrument could provide important reference information to interpret iodide-CIMS measurements from field studies.

The analysis of the number of detected isomers was done very well and it is very useful to see the comparison to other studies.

Response: We sincerely appreciate the reviewer's suggestions and feedback for the manuscript. In the general summary, the reviewer expressed some concerns about the instrument being too complicated to be widely used in its current form. We agree with the reviewer that the TAG-CIMS/FID is very complex, and we will simplify the design of the instrument in the future. However, we believe that the current form of the instrument, although complicated, is a reasonable starting point to demonstrate the values and capability of the simultaneous coupling of CIMS and FID to a TAG. The detailed responses of each comment can be found in the sections below.

Comment 1: The inclusion of the air/iodide mixed-reagent-ion technique is interesting, but does not seem to really belong here. It is not even indicated in the title of paper and does not address the motivations mentioned in the introduction. This is better suited to a separate manuscript. If this is

retained, it needs to be better explained how the mixed-reagention approach relates to the other capabilities of the instrument. An additional figure or other concrete example should be shown to demonstrate how chemical information can be derived from this technique.

Response: We thank the reviewer for thinking that it is interesting. Both reviewers highlight multi-reagent chemistry as both interesting and perhaps not fully developed. The intention of including this approach is not necessarily to understand in great detail this specific reagent chemistry, as we agree that an in-depth exploration of such an approach would likely require a separate full manuscript (note, for example, that the core of understanding of iodide chemistry was only achieved over multiple separate manuscripts). Instead, the motivation of this work is to overcome the technical hurdles in coupling a GC with a CIMS (and an FID) and explore the benefits of such an instrument.

We have chosen to include multi-reagent chemistry within this manuscript because it is specifically a feature made possible by coupling these instruments. Adding the GC enhances the ability for CIMS to explore new or simultaneous ionization chemistry by separating analytes with the retention time, yielding "clean" interpretations of complex spectra. Just as the separation provided by the GC allows known chemistries to be explored, it also enables the extension to new approaches such as multi-reagent ionization. We demonstrate that the multi-reagent ionization extends the range of chemical species that can be identified in the form of their elemental formulas, while the FID can serve as the detector for quantification. In the manuscript, we provide an example of the implementation of the multi-reagent ionization using a mixture of known chemical standards and liquid fragrance with unknown components on Line 360. In the example, we have shown that the elemental formulas of the extra compounds that can be detected in multi-reagent ionization mode. To emphasize the example of chemical identification using multi-reagent ionization, we have put the descriptions of the example in a separate paragraph of the manuscript:

"**An example of the benefit of this approach is demonstrated by the detection of compounds not accessible through iodide adduct formation; 4 times as many compounds are observed in multi-reagent ionization mode (with formulas assigned to at least half of them). For example, a known component in the sample of complex fragrance mixtures, eugenol (Peak 2 in Figure 4c), is identified in the multi-reagent ionization mode yet not detected in iodide mode. In Figure 4c, 6 other peaks are labeled that are not detected as iodide adducts, but for which formulas can be assigned using [M-H]$^-$ and [M+O$_2$]$^-$ as identifiers, 1: C$_{15}$H$_{24}$O, 3: C$_9$H$_{10}$O$_3$, 4: C$_{12}$H$_{24}$O$_2$, 5: C$_{16}$H$_{32}$O$_2$, 6: C$_{18}$H$_{34}$O$_2$, and 7: C$_{18}$H$_{36}$O$_2$. A reasonable objection to multi-reagent ionization is that the complexity of adding up signals in multiple ionization chemistry with variable sensitivities may prohibit reasonable CIMS quantification. However, using CIMS for identification of unknowns by formula or other chemical information is valuable on its own, and quantification of many components is achievable using the FID channel of this instrument. This technique is likely only useful when analytes are individually resolved (i.e., isomer resolution), as the resulting mass spectrum of the complete complex mixture would be otherwise too difficult to interpret.**"

Additionally, we have revised the title of this section 3.3 to clarify that this is an example of what the TAG-CIMS/FID can do:

"**3.3 Exploring new chemistries: multi-reagent ionization** "

Comment 2: Instead of the mixed-reagent ion description, it would be better to include an assessment of the FID-enabled determination of CIMS sensitivities. This is stated several times as the major benefit of the FID. However there are no data shown to demonstrate this utility. This is a significant weakness of the manuscript. The manuscript could be published with some minor revisions, but would have a much larger impact if the use of the FID to determine sensitivity is demonstrated.

Response: We completely agree with the reviewer that the FID-enable determination of CIMS sensitivities is a major benefit of coupling an FID and should be discussed in detail. However, quantification of iodide CIMS is a complex topic (again, note current quantification schemes have been built up over multiple manuscripts), and a proper treatment of it here would significantly expand the scope and complexity of the present manuscript. We choose instead to focus here on the technical hurdles and the array of potential value/benefits of the coupled instrument. A forthcoming manuscript is nearly ready for submission that addresses in detail the FID-determined CIMS sensitivities, and a wide range of related topics, including an examination of the voltage scan calibration method and the correlation between GC retention time and CIMS sensitivities. We have provided the information on splitting the work as described on Line 226:

"Implementation of this calibration approach including detailed methods of quantification and determination of isomer sensitivity is complex and will be addressed in future work. **This manuscript focuses instead on the descriptions of technical hurdles overcome by TAG-CIMS/FID and its potential value in understanding existing and new ionization chemistries, as well as atmospheric systems.** "

Comment 3: Specific/minor comments: Section 2.3. It would be helpful to summarize the major technical challenges of CIMS/FID split flow at the top of this section.

Response: We agree with the reviewer that summarizing the major challenges of splitting the flow at the beginning of section 2.3 would help readers to understand the section better. We have added the summary at the top of this section:

**"The design of the TAG-CIMS/FID interface needs to allow the efficient transfer of analytes from GC effluent to CIMS and FID. This interface is subject to three technical challenges:1) the connections between capillary columns and fittings need to be leak-tight; 2) all components in the interfaces require proper heating to avoid cold spots and dead volume; and 3) the relative flow rates to CIMS and FID need to be controlled to maintain roughly equal split of flow.** **"**

Comment 4: Page 6 line 166- 170. I don't understand the purpose of the EI-MS experiment. Please explain.

Response: We apologize for the lack of explanation on the EI-MS experiment. The EI-MS experiment was used as a secondary way to verify the stability of the flow split in addition to the flow rate measurement. Since the split flow rate ratio is critical for quantifying analytes in this instrument, we examined the flow split using two methods. The first method is to directly monitor the flow rate at the FID side during a

chromatographic run. Since the total flow rate is known and controlled by TAG, the remainder flow to the CIMS can be calculated. Details of the description can be found on Line 166:

"With these dimensions and temperatures, the flow rate to FID is approximately one-third of total GC flow (0.3 sccm, measured using Sensidyne Gilibrator-2 at the inlet of FID) with the remainder to the CIMS (0.7 sccm). To further evaluate the stability of the split ratio of flow, test runs were conducted prior to the experiments to monitor the flow rate at the inlet of FID, variability in the flow split was found to be less than 10% variation throughout a run cycle, stable enough to be quantitative."

Additionally, we examined the split ratio through the injection of a mixture of alkane standards (C8-C40). Suppose that the split flow ratio is constant throughout a chromatographic run. In this case, the ratio of chromatographic peak area between the two detectors should be the same for a series of n-alkanes, which elute at different retention times depending on their carbon number. However, this cannot be done with an iodide CIMS because it cannot detect alkanes. Since both CIMS and EI-MS is at near-vacuum, swapping CIMS to EI-MS, which can measure alkanes, does not impact the flow split ratio. We therefore replaced CIMS with the EI-MS to evaluate whether the flow split ratio between CIMS and FID is stable in a GC run cycle. We found that the ratios of EI-MS to FID peak area for observed alkanes, which linearly correlate with the flow split ratios at a given retention time, were found to vary by less than 10%.

The EI-MS experiment was conducted for validating the TAG-CIMS/FID interface design before actually coupling the FID to the CIMS. We thought there is no harm to provide more information on the ways of validating the flow split. However, based on the reviewer's comment, adding the extra EI-MS verification method may confuse the reader. Since the measurements of FID flow are sufficient to demonstrate a stable flow rate split, we have deleted the description on the EI-MS method used to validate the flow split:

"Additionally, the CIMS was swapped with an EI-MS while maintaining the TAG-MS/FID interface so that liquid injections of alkanes standards (i.e., alkanes mix C8-C40, AccuStandard) can be measured by both EI-MS and FID. The ratios of EI-MS to FID peak area for observed alkanes, which linearly correlate with the flow split ratios at a given retention time, were similarly found to vary by less than 10%."

Comment 5: Page 10 Line 235: Wouldn't decomposition during TD change the parent formula as well?

Response: We are not quite clear to the reviewer's question since the sentence has already discussed the decomposition of parent analyte during thermal desorption. In response to what we think the reviewer refers to, we have revised the sentence on Line 236 to mention that thermal desorption may change the parent formula:

"**Overestimation may occur when large parent molecules decompose to isomers of a smaller formula during thermal desorption.** Overestimation may occur because peaks observed might be formed in part by thermal decomposition of analytes during thermal desorption."

Comment 6: Page 11 Line 260: This statement should be qualified with the volatility- or carbon number-range analyzed.

Response: We agree with the reviewer that the conclusion needs to be further constrained on specific samples or the range of compounds. The data in the literature is collected using off-line filters.

Therefore, the compounds in the literature and our study are primarily particle-phase compounds (or some adsorbing low-volatility gases, which are known to be present on some filter samples), with ten or less carbon number. We have added this description in the manuscript on Line 260:

"Together, the published data and that collected by TAG-CIMS/FID support the conclusion that isomers are abundant **for molecular formulas with ten or less carbon number in particle-phase samples**."

Comment 7: Page 11 lines 264-266: Could you not use the EI-MS and identify the isomers via matching to a NIST library?

Response: As noted in our response above, while it is possible to include all three EI, CIMS, and FID detectors, doing so was not a focus of this work; EI data was not analyzed for the samples shown here. As addressed in Comment 4, we apologize for misleading the reviewer to believe that the EI-MS was used to measure analytes in this study.

Although not described in the manuscript, we indeed tried using the EI-MS as the 3rd detector to achieve a simultaneous coupling between the TAG and three detectors (iodide CIMS, EI-Tof-MS, and FID) for some test runs during the limonene-$O_3$ experiments. Unfortunately, even for that data, most of the oxidation products do not have reference mass spectra in the NIST library. This is a common problem for atmospheric GC/EI-MS samples. For example, the EI mass spectra of some primary limonene oxidation products (Witkowski and Gierczak, 2017) such as ketolimononic acid and 7-hydroxy limononic acid are not reported in the NIST library. A common practice of EI-MS users is to build a customized EI-MS library for specific compound categories of their interests (Yee et al., 2018), but such customized libraries can be labor-intensive and it is often the case that the analytes can only be related to a parent molecule, so it is not clear such a task would advance the scope of the present manuscript.

Comment 8: Page 14 lines 330-358: Can you provide an example of how this can be used? This seems to subvert the main benefits of CIMS, which are a 1:1 correspondence between molecule and product ion, and the retention of the parent ion (low fragmentation). If the only benefit is that it allows detection of some other compounds with the CIMS, why not just use the FID, which detects everything and with roughly equal sensitivity? The spectra are probably too complicated to interpret without GC-preseparation. Maybe there is some information about the structure of the isomer, given the observed mixed I-/O2- product ions. If so, can you provide some examples? Please comment on the use of this method to interpret the non-adduct ions typically observable in I- CIMS spectra. Can you state anything concrete about the identity of these ions, given the results of your experiments?

Response: We agree with the reviewer that the multi-reagent chemical ionization is not practical to be used in direct-air-sampling CIMS. As discussed above, multi-reagent ionization is discussed in this manuscript precisely because it is an approach that has some benefits (extending the potential chemical range of the instrument) but is only made possible with GC pre-separation. It is certainly true that the spectra are probably too complicated to interpret without GC pre-separation. The multi-reagent ionization method proposed in this study should be used mostly in the setup of the TAG-CIMS/FID (or some other GC-CIMS) where CIMS serves as an instrument for elemental formula identification and FID quantifies the well-resolved analytes.

The reviewer asked that if the only benefit is that it allows detection of some other compounds with the CIMS, why not just use the FID, which detects everything and with roughly equal sensitivity. It is because the information provided by the FID is very limited. Since FID is a single-channel detector, simply using GC-FID alone does not provide any information on the analyte molecule except its FID abundance. Additionally, the range of FID sensitivities for oxygenated organics still vary by a factor of two (Scanlon and Willis, 1985) while it can be corrected to within 20% if the elemental formulas of the analyte are known (i.e., corrections using O/C and carbon number) (Hurley et al., 2020). As mentioned in Comment 2, this FID-assisted calibration technique will be discussed in detail in a forthcoming manuscript. Simultaneous coupling of the CIMS in multi-reagent ionization mode can identify elemental formulas of chemicals generated in the atmospheric oxidation products. Those identified formulas can serve as correction factors for the FID calibration and also provide an additional dimension of the information (i.e., formula-level information) on the molecule. In this case, increasing the number of detected compounds using multi-reagent ionization in CIMS is beneficial so that more compounds can be identified and quantified.

The example on the identification of the ions was provided on Line 357 :"For example, a known component in the sample of complex fragrance mixtures, eugenol (Peak 2 in Figure 4c), is identified in the multi-reagent ionization mode yet not detected in iodide mode. In Figure 4c, 6 other peaks are labeled that are not detected as iodide adducts, but for which formulas can be assigned using $[M-H]^-$ and $[M+O_2]^-$ as identifiers, 1: $C_{15}H_{24}O$, 3: $C_9H_{10}O_3$, 4: $C_{12}H_{24}O_2$, 5: $C_{16}H_{32}O_2$, 6: $C_{18}H_{34}O_2$, and 7: $C_{18}H_{36}O_2$."

To better clarify the applicability of the multi-reagent ionization, we have revised the section by moving the example to a separate paragraph demonstrating the more formulas can be identified.

"**An example of the benefit of this approach is demonstrated by the detection of compounds not accessible through iodide adduct formation; 4 times as many compounds are observed in multi-reagent ionization mode (with formulas assigned to at least half of them). For example, a known component in the sample of complex fragrance mixtures, eugenol (Peak 2 in Figure 4c), is identified in the multi-reagent ionization mode yet not detected in iodide mode. In Figure 4c, 6 other peaks are labeled that are not detected as iodide adducts, but for which formulas can be assigned using $[M-H]^-$ and $[M+O_2]^-$ as identifiers, 1: $C_{15}H_{24}O$, 3: $C_9H_{10}O_3$, 4: $C_{12}H_{24}O_2$, 5: $C_{16}H_{32}O_2$, 6: $C_{18}H_{34}O_2$, and 7: $C_{18}H_{36}O_2$. A reasonable objection to multi-reagent ionization is that the complexity of adding up signals in multiple ionization chemistry with variable sensitivities may prohibit reasonable CIMS quantification. However, using CIMS for identification of unknowns by formula or other chemical information is valuable on its own, and quantification of many components is achievable using the FID channel of this instrument. This technique is likely only useful when analytes are individually resolved (i.e., isomer resolution), as the resulting mass spectrum of the complete complex mixture would be otherwise too difficult to interpret.**"

Comment 9: Technical corrections: Line 221: "magnitudes" -> "magnitude"

Response: We have revised the work on Line 220:

"However, comparing to the near-universal response of FID signals, the signals of iodide CIMS per unit mole of analytes may vary up to five orders of **magnitude** and highly depend on their enthalpies of binding with iodide"

Anonymous Referee #2

 Review of Bi et al., "Coupling a gas chromatograph simultaneously to a flame ionization detector and chemical ionization mass spectrometer for isomer-resolved measurements of particle-phase organic compounds"

Summary: The authors describe a new instrument configuration using a TAG column upstream of both an FID and a ToF-CIMS. The goal was to address several current measurement issues by 1) using the GC to separate isomers in the CIMS spectrum, and 2) using the FID as a means of quantifying unknown compounds in the CIMS spectrum. They also show the simultaneous use of I- and O2- as reagent ions, with the goal of expanding CIMS sensitivity to generally less oxidized compounds. Generally, this is an interesting and well written paper that attempts to address several of the critical issues with CIMS spectrum interpretation of ambient or complex data. Separation of isomers seems be useful (at least for compounds that can make it through a GC column, which will be a subset of what I- usually can measure). This technique may well be very useful (especially for simplified systems), however I don't think the authors have fully illustrated this yet. I have a major issue with some of the methods and interpretations thereof that require major revisions. Mainly, the data shown for iodide ionization shows that the ionization process is different than in typical iodide ionization CIMS setups (at least for some compounds including vanillin). This will diminish the ability to use this instrument to guide interpretation of other typical iodide CIMS measurements. I believe the authors need to address this issue (likely by showing more measurements) before making some of the conclusions drawn here. I also have questions about decomposition, and the utility of interpreting multiple reagent ion spectra.

Response: We sincerely appreciate the reviewer for carefully reviewing this manuscript and providing insights and suggestions on the weakness of the manuscript. The main issue mentioned by the reviewer is the difference of ionization chemistry between this instrument and a direct-air-sampling CIMS. We certainly recognize this concern, but believe that much of this apparent discrepancy is due not to true differences in ionization chemistry, but rather some of the details and features of ionization are made more obvious because this instrument sees the clean spectra of individual analytes due to the pre-separation of the GC as opposed to direct-air-sampling, in which all analytes are measured simultaneously. For detailed discussions and revisions in response to the reviewer's concerns, please see our response in Comment 3 and 4. The reviewer also mentions other issues on thermal decomposition of analytes, which are addressed in the responses of Comment 1 and 2; some of these concerns are due primarily to a lack of clarity in our original manuscript around operating conditions and temperatures, which we apologize for and have tried to correct in the revision. The question on the quantification using multi-reagent ionization is addressed in the response of Comment 5.

Some of the reviewer's comments involve questions of the capability of this instrument to do quantification of analytes. We would like to note that the main objective in this study is to demonstrate the detailed instrumental design for isomer-resolved measurements of particle-phase organics. Such an instrument could be useful to investigate several different issues, include quantification and novel ionization chemistries. In many cases, investigating any one of these questions in detail is likely complex, and we feel that it is best examined in separate manuscripts to give a proper full treatment. In fact, we are preparing another manuscript on the quantification and sensitivity of this instrument. The forth-coming manuscript will discuss the quantification of CIMS sensitivities, the variance of isomer

sensitivities within a formula, examination of the voltage scan calibration method, and correlation between GC retention time and CIMS sensitivities. Including such information in this manuscript would substantially expand the complexity and scope of this manuscript, which we feel would not serve readers well either in terms of understanding the present work, or understanding issues related to quantification and sensivitity. We have provided the information on splitting the work as described on Line 226:

Implementation of this calibration approach including detailed methods of quantification and determination of isomer sensitivity is complex and will be addressed in future work. **This manuscript focuses instead on the descriptions of technical hurdles overcome by TAG-CIMS/FID and its potential value in understanding existing and new ionization chemistries, as well as atmospheric systems.** "

What we would like to highlight in this manuscript is that this instrumental design provides options to investigate known and new ionization chemistries, which includes in-depth issues such as quantifying sensitivies of iodide CIMS and expanding the range of identified chemical species using multi-reagent ionization. We have added a sentence to highlight the needs of future work on Line 365:

**"We demonstrate here an example of exploring new reagent chemistries: simultaneously using multiple reagent ions is only made possible by the GC separation of analytes, but expands the information provided by this instrument. An in-depth understanding of the competition between reagent chemistries in a multi-reagent system is beyond the scope of this manuscript."**

Comment 1: Main comments:

Line 112: Is there a risk of decomposition of organic analyte molecules at this 225C temperature in the IMR, especially on metal surfaces? I would also be worried about fragmentation at the 300C temps upstream. Previous FIGAERO CIMS research has suggested decomposition of oligomers and/or highly functionalized molecules when heating at even lower temperatures than 300C. Please add some references to this previously observed issue, and also add some discussion somewhere in the manuscript of how decomposition would affect your measurements.

Response: We agree with the reviewer's concern about the potential for high temperatures to produce decomposition of target analytes, and we seek to address those concerns throughout our responses here and in the revised manuscript. The specific issue raised in this comment is actually simply a misunderstanding due to a lack of clarity in the original manuscript suggesting that the IMR temperature was held at 225 $^{\circ}$C in our study. In reality, the IMR was not heated, but instead was kept at room temperature (~20 $^{\circ}$C). The transfer line between the passivated flow splitting manifold and the IMR consisted of inert silica tubing (GC guard column) that was maintained at 225 $^{\circ}$C using a heated metal sheath and interface. A transfer line using the same materials was also used to connect the flow splitting manifold to the FID in order to ensure that both detectors "see" the same analytes. To clarify the misunderstanding, we have revised the manuscript on Line 114:

"The inlet is a heated metal **interface** which is kept at 225°C and has a 1/32" inner diameter bore-through center hole to allow insertion of fused-silica guard column into the ion-molecule region (IMR) **which is kept at room temperature (~20 $^{\circ}$C)**."

Because the heated inlet was directly connected with the IMR, the outer IMR surface facing the inlet had a slightly higher temperature (~50 °C) due to the heat transfer. However, the temperature inside the IMR was room temperature and the surface temperature (50 °C) was not substantially higher than the typical CIMS operating temperature. Additionally, the room temperature reagent ion flow rate (2 slpm) was three orders of magnitudes higher than the heated GC column flow rate (~0.7 sccm @ 225 °C), so the temperature of the mixed flow in the IMR is not expected to be significantly elevated.

Although IMR temperature is a misunderstanding due to unclear aspects of the description, we acknowledge that the thermal-decomposition of analytes can impact the interpretation of the collected data. This decomposition is expected to occur primarily during the heating of TAG sampling cell where compounds may decompose to volatilize, and in the GC column and flow splitting manifold, where compounds are exposed to higher temperatures to mobilize them. As the reviewer notes, we agree it is a concern with any instrument that uses thermal desorption that some artifacts or misinterpretation may occur due to the decomposition of thermally unstable products. Some of those limitations are discussed in the manuscript on Line 285:"thermal desorption may fragment larger accretion products to form analytes not present in the original sample (Isaacman-VanWertz et al., 2016; Lopez-Hilfiker et al., 2016b), or may reverse particle-phase oligomerization reactions (Claflin and Ziemann, 2019)."

Critically, the transfer lines to both detectors are held at temperatures at or below that of the flow splitting manifold to avoid any further decomposition. This is meant to ensure that any thermal decomposition occurs upstream of the flow split and both detectors see the same effluent mixture. In other words, while the eluting analytes may not be identical to the sampled analytes, both detectors see the same eluting analytes. We agree that scientific interpretation of these atmospheric data consequently needs to consider the possibility of decomposition adequately and be interpreted within this context, but decomposition is not expected to influence the comparison of signals between CIMS and FID. To this end, we strive to be transparent and clear about the potential impacts of decomposition on the isomer analyses. As suggested by the reviewer, we have added more references to show that compounds are subject to decomposition in the thermal desorption process and elaborated the discussion of the impacts of such decomposition on the results on Line 273:

"Conversely, thermal desorption **within TAG** may fragment larger accretion products to form analytes not present in the original sample **(Buchholz et al., 2019; Isaacman-VanWertz et al., 2016; Lopez-Hilfiker et al., 2016b; Stark et al., 2017)**, or may reverse particle-phase oligomerization reactions (Claflin and Ziemann, 2019). **These fragments may not represent the actual molecular composition of SOA, though they nevertheless may provide insight into the formation mechanisms of SOA (Isaacman-VanWertz et al. 2016). Consequently, the potential multiple fragments from one parent compound may result in an overestimation of the number of isomers.** We note, however, that similar numbers of isomers are observed when using liquid chromatography (Figure 3b), which does not involve thermal desorption. Given these uncertainties, we believe that the results presented are not a floor or a ceiling on the number of isomers in the atmosphere, but a step toward understanding a poorly constrained problem."

Comment 2: Line 211: Related to my previous comment, how much of these early eluting compounds might be fragmentation products as an artifact of the sampling technique? Fig 4: Regarding decomposition again, one way to investigate would be to show the FID signal for Fig 4 and maybe for

more examples where you are injecting single known compounds. How much signal is in the FID but not the CIMS, i.e., how much fragmentation occurs throughout the sampling process? I think this would be important information for a reader to judge the utility of the technique.

Response: We interpret this comment to raise two possible and related issues on the subject of decomposition. Firstly, decomposition of sampled analytes in the TAG instrument ("upstream decomposition") may mean that detected analytes are actually transformed products of the true sampled analytes. Secondly, if decomposition impacts each detector differently ("downstream decomposition"), comparisons between chromatograms or detectors may be biased or incorrect. We discuss both here in order to address the reviewer's concerns.

Firstly, we agree that decomposition occurring during sampling or transfer of the sample to and/or through the GC column would impact the scientific interpretation of collected data. Critically, we note that decomposition upstream of the flow splitting manifold should not impact the comparison of CIMS and FID data, which would see the same (transformed) analytes. In the context of this work, the impact of upstream decomposition within TAG primarily would impact the counting of isomers, as fragmentation may lead to overestimation of isomer counts, which we have sought to make clear in the revised manuscript on Line 236 as excerpted below. In any work that uses this or any GC-based instrument, similar considerations will need to be taken in the context of any conclusions drawn. These instruments consequently always offer/suffer some tradeoffs between molecular specificity and potential for in-instrument transformations, but their ability to measure specific tracer molecules indicative of sources and chemical pathways (whether or not those tracers are actually decomposition products) has nevertheless provided a range of important atmospheric conclusions, such as the body of work from other TAG instrumentation (Isaacman-VanWertz et al., 2016; Williams et al., 2006; Zhao et al., 2013).

Line 236: "**Overestimation may occur when large parent molecules decompose to isomers of a smaller formula during thermal desorption.** "

Secondly, we consider the possibility for downstream decomposition, such that that the early eluting compounds appeared in FID yet missing in CIMS in Figure 2 is due to the thermal decomposition of analytes in CIMS or FID. Some of this concern may again be due to the implication in the original manuscript that the IMR was heated. Instead, we have sought to design this instrument such that any decomposition occurs upstream of the flow split, as described above, so we believe decomposition is far more likely upstream of the flow splitting manifold than downstream. Specifically, we note that decomposition in transfer lines downstream of the flow splitting manifold should be minimal because the transfer lines are kept at or below the temperatures of the interface, and downstream decomposition is probably not a significant process. The compounds coming out of the TAG, which may be the fragments of a parent molecule, are therefore being measured by the CIMS and FID simultaneously.

Given these considerations, we believe the differences between FID and CIMS chromatograms can be best explained by the wide ranges in CIMS sensitivity previously reported in the literature, as described in the manuscript. While an FID provides a mass-based, near-universal response to hydrocarbons and oxygenated organics, the range of the iodide CIMS sensitivities can vary up to 6 orders of magnitudes (Iyer et al., 2016), with a general tendency to be more sensitive to polar or hydroxyl-containing

compounds. Many oxidation products of monoterpenes are less-polar compounds (e.g., limona ketone (Donahue et al., 2007)) and even compounds having a single hydroxyl group is not necessarily sensitive in an iodide CIMS. Since less-polar compounds tend to elute early on a polar GC column (i.e., MXT-WAX used in this study), those early elutes are expected to be not detected or have low abundance in the iodide CIMS. In contrast, polar compounds tend to elute late in the chromatogram and can provide orders of magnitude stronger signals in CIMS.

Overall, therefore, it is not clear that decomposition could account for the observed differences in CIMS and FID chromatograms, while known trends in sensitivity provide a reasonable explanation. Some early eluting compounds may be present in the CIMS chromatogram at low signal but not visible because displaying them in a single chromatogram is difficult with a linear Y-axis due to the wide range of sensitivity.

To explain the two reasons that result in the differences between CIMS and FID chromatograms, we have added more descriptions on Line 206:

"Since the TAG-CIMS/FID interface and the capillary to the FID is held at 50 °C above the maximum column temperature, differences in the transfer of analytes to these two detectors should be negligible. Instead, these differences are due to the selectivity of the two detectors. FID is a near-universal detector, able to detect almost all organic compounds with relatively similar and predictable responses (Scanlon and Willis, 1985). **The sensitivity of the iodide-CIMS may differ by orders of magnitude and is highest for compounds that contain multiple OH groups and can therefore more readily form an adduct with the iodide ion (Iyer et al., 2016). Since the TAG here used a polar (MXT-WAX) GC column that more preferably retains polar compounds, the early-eluting compounds are likely less-polar, and consequently less sensitive or not detected in the iodide-CIMS. Some early-eluting compounds may be present but have peaks too small to be visible due to the linear display of signal in Figure 2.** "

In the context of the reviewer's concerns, we understand their suggestion to add the FID chromatogram for Figure 4 in the manuscript to probe the decomposed fragments by comparing chromatograms between CIMS and FID. As above, the reviewer suggests that what is shown in the FID chromatogram yet missing in CIMS chromatograms may be decomposition fragments due to heating of the sampling process. We appreciate the reviewer for suggesting options to identify the thermal decomposition compounds in the instrument. For the reasons discussed above, we do not think decomposition is a likely explanation for the observed differences between chromatograms. In Figure R1, we have included the FID chromatogram for Figure 4. The sample injected is a mixture of know chemical standards and unknown fragrance liquids. As in the case of Figure 2, there are clearly more peaks in the FID, as some compounds in the fragrance are expected to be hydrocarbons or lightly-oxygenated compounds that an iodide CIMS cannot detect (e.g., monoterpenes, $C_{10}H_{16}$, a major component of fragrances (Steinemann et al., 2011)). However, it is not clear to us that including this chromatogram provides any additional information or insight than what is already evident from Figure 2, so we have chosen not to add this chromatogram to Figure 4 in the revised manuscript. Furthermore, compounds that are sufficiently stable to be introduced to the instrument are generally less likely to decompose upon thermal

desorption, so the introduction of individual standards does not provide a clear path forward to explore the possibility of upstream decomposition

[Figure]

Figure R1: Comparison of chromatograms of analyte total ion counts between CIMS using iodide ionization, CIMS using multi-reagent ionization, and FID. Top two panels are recreated from Figure 4 of the manuscript.

Comment 3: Line 297: This inset in Fig 4a indicates a major issue. In a typical iodide CIMS setup, vanillin would be sampled almost completely at the [M+I]- cluster with iodide, but you're showing that in your instrument it is predominantly sampled at the deprotonated [M-H]- . Therefore, the ionization process is different from a normal iodide CIMS. I think this is a major problem that you need to address before publishing under the pretext that your instrument can be used to help generally interpret iodide CIMS measurements. The most likely answer that I see could be that holding the IMR at 225C is causing the changes? Perhaps at those temperatures (and with metal surfaces?), vanillin becomes a gas phase acid and $C_8H_8O_3 + I- –> C_8H_7O_3- + HI$ proceeds? At line 300, you try to address the [M-H]- by stating that vanillin "produces a large number of detectable ions through reactions with other reagents in the IMR." But, there are no other reagent ions there (except NO2- maybe, but that is often present in typical iodide CIMS spectra and is therefore not the cause), so this statement is probably not accurate. So, please address this issue thoroughly. A first step could be to do the same vanillin injection but hold the IMR at room temp. The vanillin signal may smear, but does is show up at [M+I]-? Again, this seems like a major issue since you're trying to say (eg lines 372- 374) that you can use this method to investigate non-iodide clusters separated by the 'iodide valley', but you're apparently also drastically changing the ionization method.

Response: The reviewer raises several concerns, which are generally related to the ionization occurring within this instrument, and whether it resembles that of other iodide CIMS in use in the field, with a specific discussion of vanillin as an example. A critical difference between this instrument and direct-air-sampling instrumentation is the ability to collect "clean" mass spectra of individual analytes, which leads to some subtleties in how to compare to direct-air-sampling instruments. Consequently, while the reviewer raises very reasonable concerns, we believe that a lot of the apparent discrepancy comes from the fact that this instrument specifically provides an ability to see and explore the non-adduct ions, while a typical CIMS does not straightforwardly relate adduct ions to potential non-adduct counterparts. While iodide CIMS is indeed used in large part specifically for its ability to selectively study molecular ions, non-adduct ions (i.e., ions formed through pathways other than the formation of an iodide adduct) are commonly also generated in iodide CIMS instrumentation. This issue has been previously reported, with iodide adduct ions separated from non-adduct ions by a gap in mass defect by the so-called "iodide valley" shown in Figure R2 (Lee et al., 2014). It is not unexpected that any given analyte (e.g., vanillin) could produce non-adduct ions through side reactions with small impurities in flow streams (discussed further below).

[Figure]

Figure R2: High-resolution mass defect spectrum obtained during ozonolysis of α-pinene (adopted from Lee et al. (2014))

One of the significant values of the described instrument is that it specifically provides an opportunity to study ions at the other side of the iodide valley (i.e., non-adduct ions) shown as the green dots in Figure R2 and their potential relationship to adduct ions. In a direct-air-sampling iodide CIMS, all analytes generate ions simultaneously, so parent ions and non-adduct ions cannot be independently studied. Consequently, researchers frequently discount the non-adduct ions when analyzing the iodide CIMS data. With the pre-separation of the GC, we can explore those non-adduct ions that are simply ignored in a direct-air-sampling CIMS. One of the reviewer's concern is that the 225 °C IMR temperature may cause the different ionization chemistry, which is, again, confusion due to lack of clarity in our origical description of the instrument as addressed in Comment 1; in fact, the IMR was set at room temperature so it should have minor impacts on the ionization chemistry and we apologize for the confusion and have tried to clarify as described above.

To address the reviewer's specific concern with regards to vanillin, we discuss here possible reasons for observed differences, and comparison to other previously studied analytes. The reviewer suggests that in a typical iodide CIMS setup, vanillin would be sampled almost completely at the [M+I]- cluster. We agree that it is certainly possible for an instrument to be operated in such a way, but in many typical applications of CIMS, the spectra of vanillin would be sampled simultaneously with other analytes (i.e., without the pre-separation provided by GC); given the ubiquity of non-adduct ions in CIMS spectra, in such cases, the spectrum of vanillin and/or its tendency to form non-adduct ions would often not be specifically known, so direct comparisons of our individual-analyte spectra to that of a typical CIMS are primarily limited to cases where individual analytes are introduced. More broadly, we note that while a specific iodide CIMS may be tuned to optimize measurements of a specific analyte, the mass spectrum of any given instrument may be impacted by the tuning and/or the pressure of the IMR. Spectra are therefore heavily dependent on instrumental settings such as voltage settings of the atmospheric pressure inlet (API) and operating pressure, and to our knowledge, there is no consensus on a standardized operating condition of an iodide CIMS. For example, while our IMR pressure was set at 100 mbar, we are aware of other studies using 200 mbar (e.g., Isaacman-Vanwertz et al., (2018)). Indeed, even the voltage setting of the small transfer quadrupole *rf* amplitude can impart fragmenting energy into the analyte molecules and clusters.

[Figure]

Figure R3. Iodide subtracted mass spectra for liquid standards of levoglucosan, undecanoic acid, and hexdecanoic acid in this study.

To provide the reviewer with further information on the ionization scheme of this instrument, we provide the mass spectra for liquid standards of levoglucosan and undecanoic acid in this study in Figure R3. Those compounds have [M+I]- as the most abundant ion in the "clean" mass spectrum. Levoglucosan, which was reported having the near collision-limited sensitivity in Lopez-Hilfiker et al., (2016a), is also found to form mostly iodide-adduct in this study. The results suggest that the ionization chemistry varies significantly for different compounds, and that more polar (more sensitive) compounds do tend to have more dominant adduct ions. We have revised some description on the ranking of the detectable ions in the manuscript on Line 305:

"**Although the abundance ranking of the produced ions may differ on a compound-by-compound basis, we constantly observe ions other than [M+I]⁻ in the clean mass spectrum of injected liquid standards such as undecanoic acid, hexdecanoic acid, and 1,12-dodecanediol, as well as more polar**

**and low volatility aerosol constituents produced in the oxidation experiments.**  "

Overall, the TAG-CIMS has a unique advantage of exploring ionization chemistry due to the pre-separation of a GC, but, for the same reason, there are not a lot of published spectra in the literature against which to compare to other CIMS. However, the target of this study is not to investigate the ionization chemistry for specific compounds, but to propose a proof-of-concept instrument for future studies on those specific science questions.

A second question in the comment: "At line 300, you try to address the [M-H]- by stating that vanillin "produces a large number of detectable ions through reactions with other reagents in the IMR." But, there are no other reagent ions there (except NO2- maybe, but that is often present in typical iodide CIMS spectra and is therefore not the cause), so this statement is probably not accurate."

We agree with the reviewer that the statement is not accurately explained and the "other reagent ions" were not clearly defined in this sentence. In fact, there are other reagent ions with their abundance too small to be observed in the figure. Although the abundance of O2- in the iodide ionization mode is very small, 0.03% compared with I-, it may be still competitive to react with the analytes thus produced [M-H]-. The O2- is likely produced by the impurity of UHP N2 (99.999%). Since reagent ions can have dramatically different sensitivity to the analyte, the low abundance of O2- can probably not be ignored in the ionization process. We suspect that the [M-H]- is generated mainly through the reaction between the analyte molecule and the $O_2^-$ because the abundance of [M-H]- ions of most analytes were boosted five to ten times after mixing 5% of the zero-air in the ultra-high purity (UHP) $N_2$ reagent ion flow. Additionally, previous studies have reported such ionization pathway of O2- as described on Line 317:"It is reported that the presence of $O_2^-$, which is commonly found in atmospheric pressure ion sources such as electrospray ionization (ESI) (Hassan et al., 2017), atmospheric pressure chemical ionization (APCI) (McEwen and Larsen, 2009), atmospheric pressure photoionization (APPI) (Song et al., 2007), and direct analysis in real-time (DART) (Cody et al., 2005), may result in the deprotonated molecules through oxidative ionization."

Mentioned the reaction with other reagent ions before introducing the reaction mechanism probably lead to the confusion of the reviewer. To avoid such confusion, we have deleted the description on "other reagent ions" on Line 303:

**"In other words, this compound, which is generally measurable by iodide-CIMS (Gaston et al., 2016), produces a large number of detectable ions other than the iodide-adduct ions** **"**

Comment 4: Line 303: Interesting that the lower volatility compounds or more polar compounds appear to have different ionization processes relative to the more volatile or less polar compounds like vanillin. Since one of the advantages of I- ionization has been a more consistent (and single) ionization process for the majority of compounds, do you have any thoughts on how this affects interpretation of the spectra?

Response: We agree with the reviewer that one of the main benefits of the iodide CIMS is the simple ionization chemistry. The simple adduct formation makes the quantification using an iodide CIMS straightforward by only tracking the abundance of the iodide-adduct. The findings in our study do not change this fact. However, not all analyte molecules form an adduct with iodide. Previous studies have reported some iodide-adducts, such as simple monocarboxylic acids or diols, may be rapidly disassociated when increasing the voltage differences in specific components of the API (Lopez-Hilfiker et al., 2016a). Additionally, the use of multi-reagent ionization in our study demonstrates that other reagent ions can compete with iodide and produce non-adduct ions. The finding here suggests that there is not only iodide chemistry in the IMR, but also other potential ionization pathways producing the ions at the other side of the iodide valley. As noted above, we believe this to be one of the major advantages of a coupled GC-CIMS system - to better understand ionization chemistries and examine features that are otherwise difficult to study (such as non-adduct ions). To avoid the misunderstanding that iodide ionization chemistry may change with the polarity of compounds, we have revised the manuscript on Line 305:

"**Although the abundance ranking of the produced ions may differ on a compound-by-compound basis, we constantly observe ions other than [M+I]$^-$ in the clean mass spectrum of injected liquid standards such as undecanoic acid, hexadecanoic acid , and 1,12-dodecanediol, as well as more polar and low volatility aerosol constituents produced in the oxidation experiments.** "

Comment 5: Line 318: Since you're seeing the deprotonated ions using both ionization methods (just I-, and both I- and O2-), it seems like it would be really hard to convert the signal in multiple reagent ion mode to mixing ratios as the sum of two ionization processes with variable sensitivities. Especially this would be hard if you have are sampling a complex mixture. It makes me wonder if it's feasible to use both reagent ions at the same time, or if you should instead use them one at a time in series. No doubt that O2- ionization gives you a lot of extra information about the less oxidized/polar compounds that Ican't see. Please discuss this to give the reader confidence that using two reagent ions simultaneously is actually a practical scientific improvement. Best would be to show a quantitative example of a calibration curve using both reagents, but possibly this will be the subject of a future manuscript.

Response: We fully agree with the reviewer that it is complicated to calculate the mixing ratio of an analyte by summing up concentrations/abundance in the two ionization processes of CIMS without building a deeper understanding of the O2- pathway. However, this is not what we intend to do. In the multi-reagent ionization mode, the CIMS is primarily valuable for examing elemental formulas while the quantification could be achieved by FID. We agree that alternating different modes might be helpful, but note that due to the inherent semi-continous nature of GC, that would come with its own tradeoffs in terms of time resolution and the fact that each analysis might not be examining exactly the same sampled air. We have revised the sentences on Line 360 to clarify the issue:

"A reasonable objection to multi-reagent ionization is that **the complexity of adding up signals in multiple ionization chemistry with variable sensitivities may prohibit reasonable CIMS quantification.** However,

using CIMS for identification of unknowns by formula or other chemical information is valuable on its own, and quantification of many components is achievable using the FID channel of this instrument."

Comment 6: Technical comments: Fig. 3 Caption: extra 's' after SOA

Response: We have deleted the "s" in the caption of Figure 3:

"Figure 1. …… combined datasets from **SOA** formation ……"

Comment 7: Line 324: This sentence just needs some clarification. You're either adding zero air to the ionizer alongside methyl iodide, or adding O2- to the ionization region (aka IMR) alongside iodide. Also "ionization region" is ambiguous because that could be the ionizer or the IMR.

Response: We agree with the reviewer that the description on Line 324 is ambiguous. We have revised the sentence on Line 328 for clarifications :

"the CIMS was operated in a multi-reagent ionization mode by adding **100 sccm flow (i.e., 5%) of ultra-zero air to the 2 slpm flow of $N_2$ for the gas supply of the methyl iodide permeation tube**."

Comment 8: Line 326: This was confusing to me because you list the numbers with multi-reagent ion first and iodide second, right after referring to Figs. 4b (iodide) and 4d (multi) in the reversed order. Please reverse the order of listing the numbers in order to stay consistent with the Fig.

Response: We thank the reviewer for pointing out the reversed order of ionization mode. We have revised the manuscript on Line 330:

" the total ion counts are **42.4**×$10^6$ and **41.4**×$10^6$ ions/s and the I$^-$ ion counts are **1.8**×$10^6$ and **0.7**×$10^6$ ions/s for **iodide** ionization and **multi-reagent** ionization, respectively."

---

## Referee Report (RR1)

**Review of** Coupling a gas chromatograph simultaneously to a flame ionization detector and chemical ionization mass spectrometer for isomer-resolved measurements of particle-phase organic compounds
**AMT, 2020-264**
Bi et al.

**Comments:**
This is a second review of the revised manuscript. The authors have addressed most of my main concerns from the previous version.

My remaining concerns are as follows:
1) The use of the FID measurement to calibrate the instrument should be shown, at least as proof-of-concept. It does not need to be explored fully, but a figure or calculation for at least one peak needs to be included. You have almost done this in lines 230-233; can you simply add some quantitative information for Analyte 1 and Analyte 2?

2) The multi-reagent-ion chemistry is an important part of this manuscript. Can you add a mass defect plot(s), such as Figure R2, comparing the ions detected with I- to ions detected with the mixed reagent ion chemistry? This could be added to the supplement. It would be helpful to show the abundance of non-adduct ions using the standard iodide ionization scheme, and their enhancement with the mixed ion mode. I think this would also help to address some of the questions from the other reviewer.

3) It is not entirely clear which figures were replaced or changed in the revised manuscript. For example, will Figure R1 replace the Figure 2 shown in the corrected manuscript? Will Figure R3 be included somewhere in the main text or in the supplement? Please indicate.

---

## Author Response (AR2)

The authors would like to thank the reviewers for the additional feedback of the manuscript and believe the revised manuscript is much improved and addresses the reviewers' concerns. We have conducted additionally experiments as requested by Reviewer 2. We have made revisions to the manuscript according to the reviewers' comments and the extra experimental findings. The colorings of text in the reviewer response are:

- Light blue: Original reviewer comments
- Dark blue: Text added in the revision while  are the text deleted in the revised manuscript.
- Black: Original text in the submitted version of the manuscript and authors' response to the comments and others.

Note that the line number in the response is based on the revised clean-version manuscript.

Since the response includes figures from the original manuscript, support information, first response letter, and the second response letter, we use the following notations to number the figures:
Figure 1, 2,…: Figures used in the original manuscript or added to the revised manuscript.
Figure S1, S2…: Figures used in the original SI (supporting information) or added to the revised SI.
Figure R1, R2…:Figures originally used in the first response letter and reproduced here.
Figure SR1, SR2…: Figures used in the second response letter (this response).

**Reviewer 1**
Comments: This is a second review of the revised manuscript. The authors have addressed most of my main concerns from the previous version. My remaining concerns are as follows:
Comment 1: The use of the FID measurement to calibrate the instrument should be shown, at least as proof-of-concept. It does not need to be explored fully, but a figure or calculation for at least one peak needs to be included. You have almost done this in lines 230-233; can you simply add some quantitative information for Analyte 1 and Analyte 2?

Response: We thank the reviewer for providing additional feedback for the manuscript. Although the detailed quantification analysis is out of the scope of this manuscript. We agree with the reviewer that a proof-of-concept of quantification for the two analytes should be shown. We have revised the manuscript on Line 226:

"The two peaks highlighted provide an example in the variability of CIMS response: Analyte 1 has a larger FID peak area, indicating a higher mass concentration in the sample mixture than Analyte 2. However, since the CIMS peak area of Analyte 1 is lower, it must be less sensitive than Analyte 2 in an iodide CIMS. With the use of FID in addition to the CIMS detector, calibration of compounds in CIMS without using authentic standards can therefore theoretically be achieved. While implementation of this calibration approach is complex, here, we provide a proof-of-concept quantitative analysis. Hurley et al. demonstrated that the number of moles of an analyte can be calculated from its FID peak area based on a calibration response factor to hydrocarbons, with a correction for oxygenation based on the chemical formula identified by CIMS (specifically, the FID response per carbon atom relative to maximum = -0.54 O/C + 0.99, where O/C is the oxygen to carbon ratio in the target analyte, Hurley et al., 2020). By applying the calibration approach for the two analytes in the example, Analyte 1 (i.e., $C_{10}H_{14}O_3$) is found to be roughly four times more abundant than Analyte 2 (i.e., $C_9H_{14}O_3$) on a per mole basis, but appear substantially lower in its CIMS signal due to a ten times lower sensitivity.  The detailed methods of quantification and

determination of isomer sensitivity will be discussed in future work. This manuscript focuses instead on the descriptions of technical hurdles overcome by TAG-CIMS/FID and its potential value in understanding existing and new ionization chemistries, as well as atmospheric systems."

Comment 2: The multi-reagent-ion chemistry is an important part of this manuscript. Can you add a mass defect plot(s), such as Figure R2, comparing the ions detected with I- to ions detected with the mixed reagent ion chemistry? This could be added to the supplement. It would be helpful to show the abundance of non-adduct ions using the standard iodide ionization scheme, and their enhancement with the mixed ion mode. I think this would also help to address some of the questions from the other reviewer.

Response: We sincerely appreciate the reviewer for suggesting a better way to examine our data and demonstrate the merit of the multi-reagent ionization method. We agree with Reviewer 1 that the manuscript should emphasize the broad signal enhancement in all non-iodide-adduct ions through the use of multi-reagent ionization mode. The comparison of mass defect plots between multi-reagent and iodide ionization mode is a great way to demonstrate such signal enhancement. Therefore, we have removed the original Figure 5 (examination of vanillin) and add instead a comparison of mass defect plots (new Figure 5) to more broadly demonstrate the enhancement in non-iodide-adduct signals after switching from iodide to multi-reagent ionization mode. We have also revised the manuscript on Line 342-376 as excerpted below. The full context of this excerpt is Section 3.3 and excerpted fully in our response to Reviewer 2 on Page 13-16 of the response.

"To examine increases in abundance of non-adduct ions in multi-reagent ionization, all identified ions are plotted as a function of their exact mass and mass defect for iodide ionization (Figure 5a) and multi-reagent ionization (Figure 5b) with the marker area representing the background-subtracted ion abundance. Analysis is limited to only ions that exhibit a chromatographic peak about the level of detection (taken as ten times signal-to-noise in the chromatographic baseline) and with ion abundance higher than 1% of the maximum signal across both systems. The results show that despite slight decreases in their abundance, nearly all of the iodide-adduct ions (green markers within the dashed circle in Figure 5a) are still present after switching to multi-reagent ionization mode. However, signals of non-iodide-adduct ions observed in iodide ionization are enhanced significantly, even for lower-polarity compounds that exhibited non-iodide-adduct ionization pathways in iodide ionization mode. Multi-reagent ionization also generates many new non-adduct ions. While shown summarily as mass defect plots, it is important to remember that all ions are not observed simultaneously, but rather elute as chromatographic peaks comprised of some subset of ions. Figure 5 consequently demonstrates that by using multi-reagent ionization, identification of compounds with iodide adduct signals can be maintained, while additional analytes are accessed through these new ionization pathways, as demonstrated by the increase in peaks observed in Figure 4c. Enhancement of these side reactions expands formula identifications to compounds that do not strongly form iodide adducts in this instrument, due either to inherent chemical limitations (e.g., low polarity) or instrument operating conditions (e.g., adduct declustering). For example, six peaks in labeled in Figure 4c are not detected as iodide adducts, but for which formulas can be assigned using $[M-H]^-$ and $[M+O_2]^-$ as identifiers, 1: $C_{15}H_{24}O$, 2: $C_9H_{10}O_3$, 3: $C_{12}H_{24}O_2$, 4: $C_{16}H_{32}O_2$, 5: $C_{18}H_{34}O_2$, and 6: $C_{18}H_{36}O_2$. ~~In multi-reagent ionization, the three most abundant ions in the vanillin mass spectrum are deprotonation (i.e., $[M-H]^-$), the cluster with $O_2^-$ (i.e., $[M+O_2]^-$), and the deprotonated dimer (i.e., $[M_2-H]^-$). Because of the presence of oxygen in the reagent ion flow, the abundance of $[M-H]^-$ and $[M+O_2]^-$ is enhanced significantly. Though the $[M+I]^-$ is no longer observed in the spectrum, this is only due to the significant increase in other signals; the actual impact on the iodide adduct formation pathway is minor. To demonstrate, we plot the comparisons of the $[M-H]^-$ and $[M+I]^-$ of vanillin between the two ionization modes in Figure 5. The peak height of the $[M-H]^-$ ion of vanillin increases by a factor of 10, from $9.0 \times 10^4$ to $90 \times 10^4$ ions/s while the $[M+I]^-$ of vanillin reduces by a factor of only 2, from $0.58 \times 10^4$ to $0.32 \times 10^4$~~

 The results suggest that the instrument selectivity to other classes of compounds can be enlarged by bringing in $O_2^-$ as an additional reagent ion, without significantly suppressing the iodide ionization pathway. In other words, the sensitivity of compounds that tend to be ionized by $O_2^-$ or other side reactions are significantly enhanced in multi-ionization CIMS with only minor decreases in the sensitivity of compounds typically observed by an iodide-CIMS. As long as individual analytes enter the CIMS at separate times, as in the case of chromatography, combining multiple ionization chemistries can provide additional information or selectivity."

[Figure]

Figure 5. High-resolution mass defect spectrum obtained for liquid mixture samples in a) iodide ionization mode and b) multi-reagent ionization mode. The area of markers is proportional to the ion abundance.

Comment 3: It is not entirely clear which figures were replaced or changed in the revised manuscript. For example, will Figure R1 replace the Figure 2 shown in the corrected manuscript? Will Figure R3 be included somewhere in the main text or in the supplement? Please indicate.

Response: We apologize for not clarifying whether those additional figures are included in the revised manuscript. Since the response includes figures from multiple sources, we use the following notations to number figures:

Figure 1, 2,…: Figures used in the original manuscript or added to the revised manuscript.

Figure S1, S2…: Figures used in the original SI (supporting information) or added to the revised SI.

Figure R1, R2…:Figures originally used in the first response letter and reproduced here.

Figure SR1, SR2…: Figures used in the second response letter (this response).

The reviewer asked "will Figure R3 be included somewhere in the main text or in the supplement?". We will include Figure S3 (previously named as Figure R3 in the first response letter) in the SI. The reviewer asked "will Figure R1 replace the Figure 2 shown in the corrected manuscript?". We do not intend to include Figure R1 in the final manuscript. The comparison of chromatograms between CIMS and FID is already shown and discussed in Figure 2 of the manuscript, and the focus of Figure 4 is the comparison of ionization modes. Our intention in Figure R1 was to answer Reviewer 2's concern, but we do not feel that including the FID chromatogram of Figure 4 in the main text provides additional information or insight than what is already evident from Figure 2, so we have chosen not to add this chromatogram to Figure 4 in the revised manuscript.

[Figure]

Figure S3. Background-subtracted mass spectra for levoglucosan and undecanoic acid in liquid standard mixture.

[Figure]

Figure R1: Comparison of chromatograms of analyte total ion counts between CIMS using iodide ionization, CIMS using multi-reagent ionization, and FID. Top two panels are recreated from Figure 4 of the manuscript.

**Reviewer 2**

I thank the authors for their responses to my comments as "reviewer 2". However, I still find that the there is a clear issue with the Fig. 4a inset, which prevents me from having full confidence that this new instrument is functioning in the way the authors are describing it should. I understand why the authors prefer to keep their main data analysis for a subsequent manuscript, but I think that this manuscript should still be able to show consistency with typical iodide CIMS instruments, and thus give confidence that the subsequent data analysis will be as interpretable and useful as suggested.

Response: The authors would like to thank the reviewer for the additional feedback on the manuscript. As requested, we have conducted additional experiments to examine whether it is the CIMS-related issue or the TAG coupling to make the mass spectrum of vanillin in this study different from what observed in the reviewer's experiments. The new experimental results suggest that the significantly high [M-H]$^-$ in the mass spectrum of vanillin is related in part to the tuning voltages and could also be influenced by the presence of a small air leak (though the latter is not known for certain). In this case, we have removed the Figure 4 inserts and acknowledged that the CIMS tuning and/or air leaks may result in imperfect mass spectra for less polar compounds like vanillin in the manuscript. Instead of focusing on a single compound, vanillin, we have revised the manuscript to more broadly demonstrate that multi-reagent ionization can boost signals to existing non-iodide-adduct ions and maintain the presence of iodide adducts using the mass defect plots, which was the original intention of showing the vanillin spectrum. We have summarized our experimental findings together with our responses and revisions in the following section.

Comments:

Comment 1: The authors' reply to my comments about vanillin ionization are not entirely satisfying. There are numerous parts of the reply that are inaccurate or irrelevant. For instance, the authors wrote: "A critical difference between this instrument and direct-air sampling instrumentation is the ability to collect "clean" mass spectra of individual analytes,…"
It is indeed straightforward to get 'clean' spectra with a direct-air sampling iodide CIMS, by just wafting a bottle of vanillin or any other single analyte in front of a sampling inlet (impurities in the commercially available vanillin are irrelevant to this discussion). This new instrument is novel because it can produce 'clean' spectra from complex mixtures which is a nice advance (though it should still produce the same spectra as when sampling a single analyte), but that's not what my comment was about. Then the authors say:
"…we believe that a lot of the apparent discrepancy comes from the fact that this instrument specifically provides an ability to see and explore the non-adduct ions, while a typical CIMS does not straightforwardly relate adduct ions to potential non-adduct counterparts."
When sampling just vanillin from a bottle directly into the iodide CIMS like I described, you do get all of the adduct and non-adduct ions, where you can directly explore which non-adduct ions form from ionization of vanillin. Therefore the authors' statement is incorrect (while the new instrument could be useful for identifying the parents of non-adduct ions in a complex mixture, again a nice advance, that's not the issue I am addressing).

Response: We would like to apologize for not clearly describing our arguments on the TAG-CIMS's ability to obtain "clean" mass spectra. We intended to mean that a TAG-CIMS can still provide "clean" mass spectra of individual analytes from a complex sample mixture. We certainly agree with the reviewer that a direct-air-sampling iodide CIMS can see "clean" mass spectra as well if only a single compound is sampled by the CIMS. This can be done by wafting a bottle of liquid standards as suggested by the

reviewer or using permeation tubes with liquid standards filled in. To further avoid such misunderstanding, we have revised the manuscript on Line 305:

"This provides a clean mass spectrum for each chromatographically well-resolved analyte and is particularly useful when analytes are in a complex mixture (Figure S3). Consequently, this technique shows a significant advantage for understanding ionization chemistry."

Comment 2: The main issue that I still have is that when you directly sample vanillin like this with typical iodide ionization, the [M+I]- signal is 100x higher than the [M-H]-- signal (I've measured this myself before), contrary to what is shown in Fig. 4a. The authors describe various things that can affect sensitivity and declustering in a TOF mass spectrometer, e.g. voltages and IMR pressure, but mostly that will only affect whether an adduct stays an adduct or declusters back to a neutral analyte and iodide anion (or further fragmentation e.g. loss of –CO2 or –H2O or –NO2 for some compounds), it won't change the ionization pathway to enhance [M-H]--.
In other words, any properly operating iodide CIMS should always sample vanillin with [M+H]- at much higher signal than [M-H]-, and never with those reversed. Because of this, I still have to conclude that something is amiss with the ionization in your vanillin example in Fig 4. The levoglucosan and undecanoic acid examples given in Fig. R3 do look fine and are as expected, but why is vanillin different? If the answer is some sort of artifact with your instrument design, then I would say there is a potentially major issue with the utility of this instrument, because your iodide CIMS data will not be comparable to other instruments for at least a subset of compounds.
If it really is that you're operating your CIMS in some atypical configuration that gives this spectrum, then I think you need to figure out why and reconfigure to something more standard, otherwise you're negating the benefits of using an iodide CIMS by unnecessarily complicating the ionization chemistry and making it incomparable to other iodide CIMS data. I understand you're arguing that your goal in this manuscript is not to fully understand all of the ionization chemistry (that's the next paper) or to have all the answers, but I would much prefer you don't publish your paper with a Fig 4a inset that is labeled as iodide ionization but is definitely strongly influenced by something else. The purpose of this paper is to show that your new instrument works and briefly show its benefits, but Fig 4a inset tells me something is not working as intended.

That said, it could still be a simple answer. I suggested that the ionization could have changed due to the IMR temperature, but the authors have pointed out that the IMR itself is not heated. It sounds like there could possibly be a surface where the transmission line mates with the IMR that could be at least somewhat heated, but barring this, there are other options. The CIMS could be operating in some strange configuration of voltages or pressures, etc, but I find this unlikely. Is it possible that you just have a lot more O2- impurity in your iodide-only mode than is typical? Is O2 diffusing in or leaking in from your zero air source or even from room air through a leaky fitting? If 225C temps are ruled out, then an O2 leak seems most likely. I'm not sure why this wouldn't show up in your Fig R3 of levoglucosan and undecanoic acid; either O2- is very insensitive to those compounds while being very sensitive to vanillin, or the potential O2 leak was occurring only in your vanillin experiment but not your Fig R3 experiments. The last thing I can think of is that in your Fig4a, that main elution peak for which you're showing the mass spectrum is not actually only vanillin, but may be dominated by some other compound that predominantly forms that non-adduct, which seems unlikely.

In summary, the Fig. 4a inset is definitely not normal iodide ionization of vanillin, and that doesn't convince me as a reader that you have shown this instrument to be sufficiently described for this

publication and subsequent use. I don't know the answer, but I suggest the authors start by sampling vanillin directly with the iodide CIMS to verify they measure mostly [M+I]-.

Response: we again thank the reviewer for their suggestions to improve our publication. To the best of our understandings, the reviewer's main concern can be summarized into two questions: 1) did the iodide CIMS configuration itself or the coupling of a TAG cause the high [M-H]-/[M+I]- ratio for vanillin? 2) if the iodide ionization chemistry is atypical, how can the iodide CIMS data obtained in this study be comparable to other CIMS instruments?

1) Did the iodide CIMS configuration itself or the coupling of a TAG cause the high [M-H]-/[M+I]- ratio for vanillin?

For the first question, the reviewer has pointed out the mass spectrum of vanillin shown in Figure 4 inserts does not agree with previous data collected by themselves and provided methods in the comments to explore the reasons for such discrepancy. The reviewer said "the CIMS could be operating in some strange configuration of voltages or pressures, etc, but I find this unlikely. Is it possible that you just have a lot more O2- impurity in your iodide-only mode than is typical? Is O2 diffusing in or leaking in from your zero air source or even from room air through a leaky fitting?" As suggested, we have completed additional laboratory experiments to specifically examine whether it is the effect of CIMS-related parameters or the coupling of a CIMS to a TAG causes the different ionization regime of vanillin. Below, we present data showing two CIMS operating conditions that can strongly affect the $[M-H]^-/[M+I]^-$ ratio of vanillin: 1) tuning, particularly the Short Segmented Quadruple (SSQ) voltages; and 2) as suggested by the reviewer, a small leak of oxygen into the Polonium ionizer.

We find that either, or both, of these instrumental changes can account for the high $[M-H]^-/[M+I]^-$ ratio of vanillin, although not observed for some other analytes. Therefore, we agree with the reviewer that high $[M-H]^-/[M+I]^-$ ratio of vanillin could be due to the CIMS-related settings and issues, but these issues are not related to the coupling to the GC, which is a major focus of this manuscript. We believe that the coupling of a GC did not change the ionization chemistry of vanillin yet the settings of CIMS, the leak of $O_2$ in CIMS ionizer, or both did.

To examine the effects of CIMS operation on vanillin ionization, we made a non-quantitative solution of HPLC water and vanillin (Sigma-Aldrich; > 99% and used without further purification). This solution was then sublimated into a clean gas flow and injected into the inlet of the CIMS using a liquid calibration system. This provided a constant gas-phase source of vanillin on which instrument parameters could be tested. The CIMS was set up using the standard flow-tube IMR and gas-phase orifice inlet (Bertram et al., 2011). No GC coupling was used. This was a "standard" Iodide CI-TOFMS as described and used frequently in the literature.

[Figure]

Figure SR1. Abundance of [M+I]- and [M-H]- of vanillin under different CIMS voltage settings.

Figure SR1 shows the raw signals for the [M+I]- and [M-H]- ions of vanillin under the "old" tuning voltages (those used for the manuscript experiments) and a "new" tuning used for the experiments described here. By switching between the old and new voltage settings, we induce an enormous change in the amount of signal observed at m/z 279 [M+I]$^-$, while the amount of [M-H]$^-$ stays the same.

A major region for the difference in the spectra is due to electronically induced declustering of the [M+I]$^-$ adduct ion (which does not affect the non-adduct ions in the same way), which has been previously demonstrated to commonly occur for less-strongly-bound adducts at larger voltage differentials (Lopez-Hilfiker et al., 2016a). To examine this specific tuning effect, we changed the differential voltage between the SSQ and the skimmer (Figure SR2; scatter around central tendencies is simply due to transients between tunings that have not been entirely discarded). The results show that the SSQ voltages, which are determined in the tuning of a CIMS, can strongly impact the [M-H]$^-$/[M+I]$^-$ ratio of vanillin (largely due to decreases in the [M+I]$^-$ ion, as shown in Figure SR1).

[Figure]

Figure SR2. The effect of differential voltage between the SSQ and the skimmer on the $[M-H]^-/[M+I]^-$ ratio of vanillin.

[Figure]

Figure S3. Background-subtracted mass spectra for levoglucosan and undecanoic acid in liquid standard mixture.

Although we use vanillin as an example to demonstrate the possibility of multi-reagent ionization using a TAG-CIMS/FID, the instrument was not intended to specifically measure vanillin. As shown in Figure S3, there are other compounds that have reasonable mass spectra in iodide ionization mode. The use of an instrument tuning that was not optimized to preserve adduct ions of weakly-bound adducts is unfortunate, and not something we would have used if we were trying to extract useful information about the atmosphere or other system, but does not impact our investigation of the GC coupling and our demonstration that GC helps clarify a complex mixture when analyzing with a CIMS. The experimental results suggest the discrepancy of vanillin ionization chemistry between our data and the reviewer's data is not due to the coupling of a GC, but the CIMS-specific parameters, which may vary depending on the application of the users.

Finally, the reviewer suggested the possibility of a leak into the ionizer. We acknowledge this could have happened during our experiments and we tested the effects of a theoretical leak by loosening the connection to the Polonium ionizer and letting a few seconds of room air into the reagent ion system. As shown in Figure SR3, we observe an enormous enhancement in the [M-H]$^-$ signal at m/z 151, relative to the system without the $O_2$ leak. As the reviewer offers, it is possible that vanillin is unusually susceptible to ionization via $O_2$- or $CO_3$-. Though we note that we do not observe a large peak for the carbonate ion in the mass spectra during these experiments, which would have been an indicator of a large leak, so it is not possible to say with certainty whether such a leak existed in the manuscript experiments.

[Figure]

Figure SR3. Normalized mass spectrum of vanillin with and without $O_2$ leak at the ionizier.

From these experiments, we conclude that the vanillin spectrum of concern to the reviewer is (a) definitely due in part to the tuning of the CIMS, and (b) could be due in part to a small leak at the ionizer (though the support for the latter is a bit mixed and is not known for certain). We hope that this satisfies the reviewer's concerns that the vanillin spectrum shown is in some way a result for the GC coupling and experiments that are the focus of this work. Further work needs to be done to probe the ionization chemistry for compounds with different functional groups, but we hypothesize that less polar compounds are more susceptible to the change of SSQ voltages and $O_2$ leak due to their weaker iodide-adduct binding enthalpies. Because tuning and physical setup does not change between iodide ionization and multi-reagent ionization, the increase in non-adduct ions observed in the latter is also not due in any way to these issues. We acknowledge the reviewer's concern regarding the vanillin spectrum and have removed Figure 4 inserts from the manuscript. We have also discussed on Line 318 and Line 400 that low signal of the iodide adduct for low-polarity compounds may be due to tuning-related declustering and/or other operating conditions.

On Line 318: "The CIMS voltage settings used in this study were not optimized to minimize declustering of lower-polarity compounds like vanillin, leading to spectra of these compounds in which the iodide adduct significantly is less dominant than the deprotonated form in, even in iodide ionization mode."

On Line 400: "While the iodide-adduct ions do exist in the mass spectrum of individual analytes, we also observe high abundance of non-adduct ions such as [M-H]⁻ and [M+O₂]⁻. Although such high abundance of [M-H]⁻ may be partially resulted by the tuning-driven declustering of low-polarity adduct ions, the observed non-adduct ions likely account for many ions in the non-adduct region of the iodide valley. "

2) If the iodide ionization chemistry is atypical, how can the iodide CIMS data obtained in this study be comparable to other CIMS instruments?

Firstly, we would like to note that CIMS-related parameters such as voltage tuning and ion-molecule reaction region (IMR) pressure can vary depending on the application of each study and there is no consensus on a constant and standard sets of voltages used in an iodide CIMS. Ideally, the signal of [M+I]-, which is typically used for quantification in an iodide CIMS, should be tuned to be as high as possible to ensure maximum sensitivity yet maintain good mass resolution. It is clearly not the case for vanillin in this study. However, although we used vanillin in the original manuscript as an example to demonstrate the possibility of multi-reagent ionization using a TAG-CIMS/FID, the purpose of the study is not to solely investigate vanillin ionization chemistry or quantify vanillin. Other compounds should be considered as well. In the iodide ionization mode, we do observe liquid standards (shown in Figure S3) that have [M+I]- as the dominant ion in their mass spectra. A further screening for all 512 compounds identified in the limonene-O₃, limonene-OH, TMB-OH, and eucalyptol-OH experiments shows that the clean mass spectra of those oxidation products have [M+I]- as the dominant ion. Therefore, we agree with the reviewer that the iodide CIMS used in this study might be less sensitive to vanillin and maybe other structurally similar compounds compared to other iodide CIMS that are tuned differently, but do not believe that unoptimized tuning towards a certain category of compounds will lead to incorrect interpretation of all data.

The original purpose of including a "close-up" of vanillin was to explore the effects and potential value of multi-reagent ionization. To address the reviewer's concerns regarding vanillin, and more broadly make this point, we have removed the original Figure 5 (examination of vanillin) and add instead a comparison of mass defect plots (new Figure 5) to more broadly demonstrate the enhancement in non-iodide-adduct signals after switching from iodide to multi-reagent ionization mode. To help the reviewer better examine the new approach, we include below the entire revised Section 3.3 in the response with marked revisions.

"3.3 Exploring new chemistries: multi-reagent ionization

[revised manuscript text omitted]

Comment 3: Additionally, about my comment 2 on decomposition products, I thank the authors for the thorough description. I intended to refer to 'upstream decomposition', where heating for desorption (and in the column) could fragment to form smaller product molecules that get transmitted to both the CIMS

and the FID. Then what I'm suggesting is that the iodide CIMS is simply insensitive to these smaller fragmentation products generally, while they do get sampled in the FID. It would be interesting (and valuable for interpretation) to know how much of this signal that gets measured in the FID but not CIMS is due to fragmentation (versus compounds that don't fragment but are not sensitive to iodide). Thank you for showing the FID signal in Fig. R1, but in hindsight that's not actually very useful because that is a mix of several/many known and unknown compounds, so there's still no way of separating out the fragmentation pathway. To answer that question, you'd have to reproduce Fig. R1, but with injections of only a single known compound/isomer at a time. Then, you should have only a single eluting peak that is the known compound/isomer, and any other peaks are fragmentation products that may well show up in just the FID and not the iodide CIMS. I think that including this type of analysis would greatly help the quality of this manuscript, but perhaps it could be included in a future manuscript instead.

Response: We thank the reviewer for clarifying that the comment is based on 'upstream decomposition'. It is true that upstream decomposition may occur when analytes were thermally desorbed from the sampling cell to the GC column. However, we note all 'upstream decomposition' is associated with TAG itself and potentially associated with all instruments using thermal desorption as the sample injection technique. Therefore, the coupling of the CIMS and FID to a TAG is not the cause of such thermal decompostion. We note that the 'upstream decomposition' issue has been reported in earlier work on TAG (Isaacman et al., 2014) and on other instruments like a FIGAERO-CIMS (Stark et al., 2017). Therefore, we agree with the reviewer that thermal decomposition related bias is an important topic and should be carefully examined.

The reviewer's suggestion would be an interesting case study to examine whether the decomposition products of one analyte are observable by iodide ionization or not, and thus might provide one possible explanation for some of the peaks. We note, though, that it is difficult to extend any such examination of one analyte to a general conclusion. For example, one could introduce a single compound that does not decompose in a GC but is ionized by I- (e.g., hexadecanoic acid) and conclude that there is no fragementation, or one could introduce something likely to decompose (e.g., a dimer) and conclude decomposition occurs and, depending on the decomposition products, may or may not be yield the FID/CIMS discrepancy observed here. A large fraction of commercially available standards unfortunately fall into the first category (do not decompose); for example, Hurley et al. introduced ~90 compounds, many of them individually, across a range of O/C up to 1.0 and showed that the peak that came out of the GC corresponded well to the analyte that went into the GC (e.g., did not appear to decompose) (Hurley et al., 2020). Conversely, Isaacman-VanWertz et al. (same PI, and PI of the present work) observed likely effects of decomposition impacting interpretation of results for ambient particles (Isaacman-VanWertz et al., 2016). Consequently, injection of individual analytes provides an exploration of decomposition for that analyte and might hint at some general possibilities, but cannot actually provide us with definitive information on the subject. We agree it would be interesting to find examples of these cases, and will consider doing so in future experiments.

As suggested by the reviewer, we have revised the manuscript on Line 215:

"It is also possible that some of those peaks are thermally decomposed analytes which exhibit low sensitivity in CIMS since all thermal desorption instruments, including the TAG, are known to potentially cause thermal decomposition of samples (Isaacman-VanWertz et al., 2016; Stark et al., 2017)"

We have also revised the manuscript based on a similar comment from Reviewer 2 in the first-round response. Here, we highlight those revisions (not shown in the track-change version of the manuscript since they were already included in the first-round revision) on Line 283:

[revised manuscript text omitted]